# Spiking Neuron as Discrete Gating for Long-Term Memory Tasks

## Abstract

Efficient long-term memory is essential for improving sample efficiency in partially observable reinforcement learning. In memory-based RL approaches, long-term memory maintenance depends critically on the sequence models used in the agent's architecture. While both linear recurrence and gating mechanisms have demonstrated improved memory efficiency, their underlying operational principles remain insufficiently explained, and their ability to filter out task-irrelevant noise remains limited. To address these limitations, we develop a saliency-based framework that elucidates how these designs enhance memory maintenance and establishes the necessity of discrete gating for effective noise suppression. Building on these insights, we design a simple yet effective recurrent cell that incorporates discrete gating through spiking neurons. Extensive evaluation demonstrates the effectiveness of our approach on Passive Visual Match, a classic long-term memory task, as well as on several other partially observable tasks with diverse characteristics. The code is provided in the supplementary material and will be made publicly available.

## 1 Introduction

Long-term memory is a critical cognitive ability that enables humans to make complex decisions by recalling and utilizing historical information in dynamic environments Kumaran et al. (2016). A key aspect of this ability is the capacity to prevent memory from being interfered with by irrelevant events during memory-intensive tasks McNab & Klingberg (2008). For Reinforcement Learning (RL) agents, achieving efficient memory, both updating essential information or memory over time and maintaining it for decision-making, is crucial for applications in real-world scenarios. The former allows agents to handle complex inputs, while the latter supports performance in long-term tasks.

Recurrent Neural Network (RNN)-based sequence models Parisotto et al. (2020); Zhao et al. (2023); Morad et al. (2023a); Le et al. (2025) offer advantages in computational and memory efficiency when handling temporal tasks, making them well-suited for real-world applications involving extended temporal contexts, such as autonomous driving Sallab et al. (2017). In this paper, we focus on the problem of **efficient memory maintenance** in RNN-based models for long-term memory RL tasks Ni et al. (2023), as shown in Figure 1 (Left).

Existing RNN-based models typically employ two key mechanisms to maintain information: linear recurrence, which enhances long-term dependency modeling Orvieto et al. (2023), and gating Hochreiter & Schmidhuber (1997); Chung et al. (2014), which explicitly selects relevant information for working memory Chaudhari et al. (2021); Morad et al. (2023a). While these mechanisms aid memory maintenance to some extent, we demonstrate that they are insufficient to fully prevent memory from being corrupted by irrelevant information (see Section 3).

In this work, we introduce a unified framework based on the signal-to-noise ratio (SNR) of temporal saliency Ismail et al. (2019) to evaluate memory maintenance in sequence models from the perspective of gradient-based backpropagation. This framework not only explains the necessity of linear and gating mechanisms, but also reveals why they fall short in achieving effective long-term memory maintenance. To address this limitation, we propose a discrete gating mechanism and design a linear recurrent cell incorporating such gating, implemented using spiking neurons Fang et al. (2023). Extensive experiments on long-term memory tasks demonstrate that our approach outperforms several state-of-the-art methods. Notably, even with a simple cell design that does not specifically optimize memory updating, our method achieves competitive performance compared to existing SOTA models

Figure 1: **Left:** Illustration of the Passive Visual Match task Hung et al. (2019) (images from Ni et al. (2023)). The task consists of input, noise, and output phases. The agent must remember the color from the input phase ($P_{in}$) and use it for color matching in the output phase ($P_{out}$). In such tasks, memory updating is straightforward during $P_{in}$ and memory maintenance is crucial. The noise phase ($P_{noise}$) acts as an interference task and can be arbitrarily long. **Right:** A typical RNN-based model unrolled over time, facilitating the analysis of memory maintenance corresponding to each phase.

on tasks involving complex inputs and memory updating requirements. Our contributions are summarized as follows:

- We propose a gradient-based analytical framework that employs temporal saliency SNR to evaluate memory maintenance in RNN-based sequence models, and identify that their inefficiency primarily arises from inadequate suppression of irrelevant information.
- We demonstrate the effectiveness of discrete gating for memory maintenance and design a linear recurrent cell integrated with a spiking neuron-based discrete gating mechanism, which enhances long-term memory maintenance while enabling parallel training.
- We empirically validate the theoretical insights derived from our framework and show the competitive performance of our model across long-term and general memory tasks.

## 2 BACKGROUND

### 2.1 MEMORY-BASED RL

Memory-based RL is proposed to solve Partially Observable Markov Decision Process (POMDP) Åström (1965), which is described in Appendix B. Early approaches are based on belief states Kaelbling et al. (1998), which require a lot of computations. Recently, model-free methods have been verified as a general and efficient approach Hausknecht & Stone (2015); Ni et al. (2022). As shown in Figure 1 (Right), the agent uses an MLP encoder and a sequence model to map the trajectory $\tau_t$ to a latent Markov state $\hat{s}_t$. Empirical studies Morad et al. (2023b) have found that traditional RNNs Hochreiter & Schmidhuber (1997); Chung et al. (2014) are sufficiently useful in simple POMDP environments. Recently, an increasing amount of research has focused on the use of modern sequence models to tackle more challenging POMDP tasks. Some methods use Transformer Ni et al. (2023), State Space Models (SSMs) Lu et al. (2023); Huang et al. (2024); Dai et al. (2024); Wang et al. (2025a), or linear RNNs Lu et al. (2024). Some methods modify the structure of linear RNNs Morad et al. (2023a) or Memory-Augmented Neural Networks (MANN) Le et al. (2025) to make them perform better in POMDP tasks. Some studies introduce more efficient training algorithms Morad et al. (2024); Elelimy et al. (2024) for certain types of sequence models. In this paper, we focus on RNN-based sequence model for Memory-Based RL.

### 2.2 RNN-BASED MODELS

Recurrent Neural Networks (RNNs) are designed to capture temporal dependencies in sequential data. However, early RNNs Elman (1990), trained via Backpropagation Through Time (BPTT), are known to suffer from the vanishing gradient problem Hochreiter (1998), which limits their capacity to model long-range dependencies and maintain memory effectively over time. Inspired by **gating**

**mechanisms** found in the human brain Gisiger & Boukadoum (2011), which regulate information flow, gated RNNs Hochreiter & Schmidhuber (1997); Chung et al. (2014); Lei et al. (2018) were introduced to mitigate gradient vanishing, thereby improving long-term dependency learning and memory retention. Despite these advances, the inherent sequential nature of such models results in $O(T)$ time complexity, slowing down training and limiting scalability to very long sequences. **Linear RNNs** Orvieto et al. (2023); Feng et al. (2024) and State Space Models (SSMs) Smith et al. (2023); Gu & Dao (2024); Dao & Gu (2024) have been developed to alleviate gradient decay and enhance long-term memory. These architectures remove non-linearities within the recurrence. The resulting linear recurrence enables parallel training through causally masked convolution or associative scan algorithms. Leveraging the complementary strengths of gating and linear recurrence, this paper proposes a unified framework based on temporal saliency analysis Ismail et al. (2019). We systematically compare the two paradigms in long-term memory tasks, revealing their respective advantages and limitations, and highlighting the critical role of discrete gating for long-term memory maintenance.

### 2.3 DISCRETE MODELS AND SELECTION MECHANISM

**Discrete models.** Discrete models use 0 or 1 to represent the data itself. Works related to discrete models include Binary Neural Networks (BNNs) Qin et al. (2020) and Spiking Neural Networks (SNNs) Roy et al. (2019), which employ the Straight-Through Estimator (STE) or Surrogate Gradients (SG) to facilitate gradient-based learning. These discrete models have many applications, such as image classification Rathi & Roy (2020); Fang et al. (2021b); Yao et al. (2022), sequence modeling Fang et al. (2023); Li et al. (2024); Chen et al. (2024), and RL Chen et al. (2022); Zhang et al. (2024); Qin et al. (2023; 2025). Their primary advantage lies in reducing computational costs, and they also serve as foundational model architectures for the discrete selective mechanism.

**Discrete selection mechanism.** The discrete selection mechanism uses 0 or 1 to decide whether to retain or discard certain information. While this mechanism already has many applications, there may be additional, yet-to-be-explored benefits. Sequence models such as SkipRNN Campos et al. (2018) and Phased LSTM Neil et al. (2016) utilize discrete selection mechanisms, significantly reducing network computation while achieving superior performance compared to traditional RNNs on supervised tasks. In other domains, Spiking-NeRF Liao et al. (2024) leverages spiking neurons, the fundamental units of SNNs, as an information selection mechanism for NeRF Mildenhall et al. (2021), functioning similarly to a gating mechanism that forms a discontinuous representation space and filters out irrelevant information.

However, almost no prior work has investigated whether discrete models or discrete selection mechanisms can improve memory maintenance by effectively filter out irrelevant information in RNN-based RL tasks. In contrast, our method explores discrete gating in such tasks. A detailed comparison with existing works across different aspects is provided in Appendix D.

## 3 LONG-TERM MEMORY MAINTENANCE ANALYSIS

**Problem Definition.** According to Hung et al. (2019), timesteps in a long-term memory task can be categorized into three distinct phases: the input phase ($P_{in}$), the noise phase ($P_{noise}$), and the output phase ($P_{out}$). During $P_{in}$, the agent acquires essential information that must be maintained in memory, even though actions taken in this phase do not yield immediate reward feedback from the environment. In $P_{out}$, the agent must retrieve the information maintained since $P_{in}$ to derive the optimal policy. The $P_{noise}$ phase consists of timesteps containing observations that are irrelevant for inferring actions or values in $P_{out}$, and the number of such timesteps can be arbitrarily large. These observations can be regarded as noisy inputs within the memory task. Nevertheless, certain memory-agnostic tasks may still occur during $P_{noise}$. To simplify our analysis, we assume that essential information or memory can be updated readily, as exemplified by tasks such as Passive Visual Match Hung et al. (2019), illustrated in Figure 1(Left). This assumption allows us to mitigate the influence of memory updating mechanisms and concentrate specifically on memory maintenance.

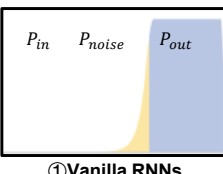 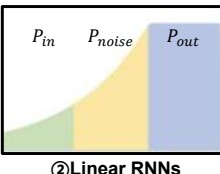 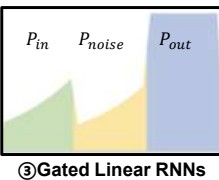 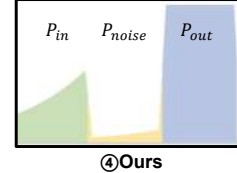

Figure 2: We present results from the Passive Visual Match task Hung et al. (2019), comparing the temporal saliency $R_t$ (Equation 3) across several RNN-based models, including a vanilla RNN Elman (1990), a linear RNN Orvieto et al. (2023), a gated linear RNN Morad et al. (2023a), and our method. In each subfigure, the horizontal axis denotes the temporal progression of the task, while the vertical axis quantifies the value of $R_t$.

### 3.1 Unified Framework for Memory Maintenance Analysis

The value network plays a crucial role in deep reinforcement learning algorithms Schulman et al. (2017); Haarnoja et al. (2018); Fujimoto et al. (2018). In our framework, its output $Q(\hat{s}_t, a_t)$ is used to assess memory maintenance. Specifically, effective memory maintenance should ensure that the contribution of observations from $P_{in}$ to the value $Q(\hat{s}_t, a_t), t \in P_{out}$ (denoted as $\mathcal{C}_{in \to out}$) is substantially greater than the contribution from observations in $P_{noise}$ to $Q(\hat{s}_t, a_t), t \in P_{out}$ (denoted as $\mathcal{C}_{noise \to out}$). Based on this concept, we define the signal-to-noise ratio (SNR) of temporal integration of contribution to evaluate long-term memory effectiveness:

$$\text{SNR}_{\mathcal{C}} = \frac{\mathcal{C}_{in \to out}}{\mathcal{C}_{in \to out} + \mathcal{C}_{noise \to out}}, \mathcal{C}_{in \to out} \geq 0, \mathcal{C}_{noise \to out} \geq 0. \tag{1}$$

The value of $\text{SNR}_{\mathcal{C}}$ lies in the interval $(0, 1)$ and reflects the relative importance of essential memory information over irrelevant one. A higher $\text{SNR}_{\mathcal{C}}$ indicates that the agent is better at maintaining task-relevant information from $P_{in}$ while suppressing interference from $P_{noise}$. To improve memory maintenance efficiency, one can therefore either **increase $\mathcal{C}_{in \to out}$** or **decrease $\mathcal{C}_{noise \to out}$**. We quantify these terms by aggregating temporal saliency values Ismail et al. (2019) across their respective phases:

$$\mathcal{C}_{in \to out} = \frac{1}{|P_{in}|} \sum_{t \in P_{in}} R_t, \mathcal{C}_{noise \to out} = \frac{1}{|P_{noise}|} \sum_{t \in P_{noise}} R_t. \tag{2}$$

Temporal saliency Ismail et al. (2019) $R_t$ at timestep $t$ is defined as the partial derivative of $Q(\hat{s}_i, a_i), i \in P_{out}$ to observation $o_t$:

$$R_t = \Big| \sum_{i \in P_{out}} \frac{\partial Q(\hat{s}_i, a_i)}{\partial o_t} \Big| = \Big| \sum_{i \in P_{out}} \Big[ \frac{\partial Q(\hat{s}_i, a_i)}{\partial \hat{s}_i} \frac{\partial \hat{s}_i}{\partial h_i} \underbrace{\Big( \prod_{j=t+1}^{i} \frac{\partial h_j}{\partial h_{j-1}} \Big) \frac{\partial h_t}{\partial x_t} \frac{\partial x_t}{\partial o_t}}_{G_t} \Big] \Big|, \tag{3}$$

where $\hat{s}$, $h$, and $x_t$ denote the output of the sequence model, its hidden state, and the encoded feature of $o_t$, respectively. Here, $R_t$ quantifies the contribution of $o_t$ to the model's output during $P_{out}$, while $G_t$ is a term that depends on the sequence model's architecture. This formulation enables a direct connection between the model structure and $\text{SNR}_{\mathcal{C}}$.

Although temporal saliency Ismail et al. (2019) has been used for memory introspection in POMDPs Wang et al. (2025b), its application has remained qualitative and lacks a phase-based analysis. We overcome this limitation with $\text{SNR}_{\mathcal{C}}$, a novel metric that explicitly distinguishes between input, noise, and output phases. This allows for direct, quantitative comparison of model architectures. Derived from BPTT gradients, $\text{SNR}_{\mathcal{C}}$ and temporal saliency both inherently reflect the efficiency of essential information propagation over time, thus linking model architecture directly to its long-term memory maintenance capability.

## 3.2 LONG-TERM MEMORY EFFICIENCY ANALYSIS FOR DIFFERENT RNN-BASED MODELS

In this section, we will show how existing sequence models with linear recurrence or gating mechanisms essentially devise a strategy to increase $\text{SNR}_\mathcal{C}$ by either increasing $\mathcal{C}_{in \to out}$ or decreasing $\mathcal{C}_{noise \to out}$ with their specific gradient dynamics.

**Non-linear RNNs vs Linear RNNs.** A unified formulation of $h_t$ in Non-linear and Linear RNNs can be formulated as:

$$h_t = f(\mathbf{W}_h h_{t-1} + \mathbf{W}_x x_t), \quad h_t, x_t \in \mathbb{R}^{N_h}, \tag{4}$$

where $x_t$ is the input of the RNN and $N_h$ is the hidden size.

In non-linear RNNs, such as the vanilla RNN Elman (1990), the recurrence typically employs a nonlinear activation function $f(\cdot) = \tanh(\cdot)$. Here, $\mathbf{W}x \in \mathbb{R}^{N_x \times N_h}$ and $\mathbf{W}h \in \mathbb{R}^{N_h \times N_h}$ are learnable weight matrices. Due to the repeated application of the nonlinearity in the recurrent step, the product term $\prod j = t+1^i \frac{\partial h_j}{\partial h_{j-1}}$ in the temporal saliency $R_t$ (Equation 3) tends to decay rapidly as $|i - t|$ increases. This leads to the vanishing gradient problem Hochreiter (1998), which in turn implies that when input-phase timesteps $P_{in}$ lie far in the past, the resulting contribution measure $\mathcal{C}_{in \to out} = \frac{1}{|P in|} \sum_{t \in P_{in}} R_t$ becomes relatively small.

By contrast, linear RNNs like the LRU Orvieto et al. (2023); Gu et al. (2020) replace the nonlinear transformation with an identity mapping $f(\cdot)$ and substitute the dense recurrence $\mathbf{W}hh_{t-1}$ with $c_t \odot h_{t-1}$, where $c_t \in \mathbb{R}^{N_h}$ is a gating vector and $\odot$ denotes element-wise multiplication. This linear structure causes the product $\prod_{j=t+1}^{i} \frac{\partial h_j}{\partial h_{j-1}}$ to decay more slowly over long time lags, enabling gradients to propagate effectively across distant timesteps. Consequently, in long-term memory tasks where $P_{in}$ and $P_{out}$ are widely separated, linear RNNs yield a larger $\mathcal{C}_{in \to out}$ (Equation 2) than their nonlinear counterparts, thereby increasing the resulting $\text{SNR}\mathcal{C}$ (Equation 1), as demonstrated by the slower decay of contribution over time in linear RNNs compared to vanilla RNNs in Figure 2 ①②.

**Non-gated RNNs vs Gated RNNs.** While linear recurrence boosts $\mathcal{C}_{in \to out}$ by sustaining $R_t$, it cannot suppress $\mathcal{C}_{noise \to out}$ to filter out $P_{noise}$. Gated RNNs Morad et al. (2023a) address this limitation by actively reducing $\mathcal{C}_{noise \to out}$, thereby enhancing overall efficiency. To this end, we formulate a unified recurrence for both gated and non-gated RNNs as:

$$h_t = c_t \odot h_{t-1} + (\mathbf{W}_x x_t) \odot \phi.$$

$$\phi = \begin{cases} 1, & \text{non-gated}, \\ \sigma(\mathbf{W}_i x_t), & \text{gated}, \end{cases} \tag{5}$$

where $\sigma(\cdot)$ is the sigmoid function, defined as $\sigma(x) = \frac{1}{1-e^{-x}}, x \in \mathbb{R}$. $\mathbf{W}_i \in \mathbb{R}^{N_x \times N_h}$ is the linear layer of the input gate. The partial derivative of $h_t$ with respect to $x_t$ in Equation 3 can be written as

$$\frac{\partial h_t}{\partial x_t} = \begin{cases} \text{diag}\left(\frac{\partial h_t}{\partial u_t}\right)\frac{\partial u_t}{\partial x_t}, & \text{non-gated}, \\ \text{diag}\left(\sigma(g_t) \odot \frac{\partial h_t}{\partial u_t}\right)\frac{\partial u_t}{\partial x_t} + \text{diag}\left(u_t \odot \frac{\partial h_t}{\partial \sigma(g_t)} \odot \sigma'(g_t)\right)\frac{\partial g_t}{\partial x_t}, & \text{gated}, \end{cases} \tag{6}$$

where $u_t = \mathbf{W}_x x_t$, $g_t = \mathbf{W}_i x_t$.

In the non-gated case, the term $\frac{\partial h_t}{\partial x_t}$ is numerically equivalent to $\mathbf{W}_x$. Consequently, provided that $\mathbf{W}x$ is a non-zero matrix, the resulting $R_t$ in Equation 3 will also be non-zero (assuming all other terms are non-zero). Since a non-zero $\mathbf{W}x$ is necessary for the model to learn valuable information during $P_{in}$, a limitation of non-gated linear RNNs is their inherent inability to reduce $\mathcal{C}_{noise \to out}$.

In contrast, the gated architecture addresses this issue. For timesteps $t \in P_{noise}$, the gating function can output a near-zero value to actively filter out irrelevant input. This forces both red terms in Equation 6 to become small, thereby making $R_t$ in Equation 3 sufficiently small. This effect is evidenced by the abrupt shifts in saliency across phases in Figure 2 ③, which starkly contrast with the smooth, gradual evolution seen in non-gated models (Figure 2 ①). As a result, $\mathcal{C}_{noise \to out}$ is effectively decreased, leading to a higher $\text{SNR}_\mathcal{C}$. This behavior can be formally summarized by the following proposition.

**Proposition 1.** *If $c_t$ is input-independent, and $\exists \boldsymbol{W}_i \in \mathbb{R}^{N_x \times N_h}$ so that $\sigma(\boldsymbol{W}_i x_t) = 0, t \in P_{noise}$ in a certain environment, $SNR_\mathcal{C}$ can reach its maximum value 1.*

A detailed proof of Proposition 1 can be found in Appendix C.1.

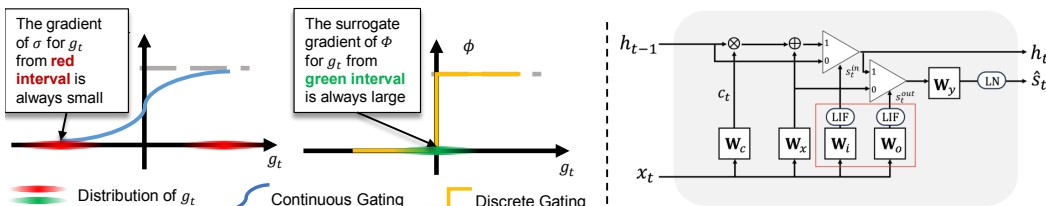

Figure 3: **Left:** Comparison of continuous Gating and discrete Gating. **Right:** Visualization of the structure of our proposed recurrent cell.

## 4 PROPOSED METHOD

### 4.1 SPIKING NEURON AS DISCRETE GATING

While sigmoid-based continuous gating can increase $\text{SNR}_\mathcal{C}$ by reducing $\mathcal{C}_{noise \to out}$, its effectiveness is limited. Due to common parameter initialization strategies and the inherent softness of the sigmoid function, the gate outputs cannot reach saturated values (*i.e.*, 0 or 1) at the beginning of training.

Even after sufficient training, when a sigmoid-based gate does eventually saturate, it encounters a different issue: gradient vanishing. Saturated outputs require gate inputs to reach extreme magnitudes, at which point the gradient of the sigmoid function, $\sigma'(x) = \sigma(x)(1 - \sigma(x))$, becomes nearly zero, as illustrated in Figure 3 (Left). This creates a fundamental dilemma, as noted in Gu et al. (2020). Consequently, sigmoid gating can lead to vanishing gradients, which diminishes the efficiency of memory maintenance.

In contrast, a discrete gating function $\phi$ trained with a surrogate gradient can produce saturated outputs without requiring its inputs to be extremely large, thereby avoiding the problem of gradient vanishing. As shown in Figure 3 (Left), when using the Heaviside step function $\Theta(\cdot)$ as the discrete gate, *i.e.*, $\mathbf{s}t = \Theta(f(x, \mathbf{W}i))$, the parameters $\mathbf{W}i$ can satisfy the conditions of Proposition 1 without needing excessively large values. This enables the minimization of $\mathcal{C}_{noise \to out}$ and the maximization of $\text{SNR}_\mathcal{C}$ from the early stages of training. As demonstrated in Figure 2 ④, the proposed approach successfully mitigates the impact of the noise phase, leading to more effective memory maintenance.

### 4.2 SIMPLE RECURRENT UNITS WITH DISCRETE GATING

**Parallel Spiking Neuron.** In POMDP tasks where inputs are partially observable, the gating function should be history-dependent so that it can control the selection mechanism based on more information. Therefore, we could use spiking neurons with temporal dynamics as the gating function. To maintain the benefit of linear RNN, which is parallel training, we use Parallel Spiking Neuron (PSN) Fang et al. (2023) as our discrete gating function, which is

$$\mathbf{m}_t = (1 - \frac{1}{\tau_m})\mathbf{m}_{t-1} + \frac{1}{\tau_m}\mathbf{u}_t, \quad \mathbf{m}_t, \mathbf{u}_t \in \mathbb{R}^{N_h},$$
$$\mathbf{s}_t = \Theta(\mathbf{m}_t - V_{th}), \qquad\qquad \mathbf{s} \in \{0, 1\}^{N_h}, \tag{7}$$

where $V_{th}$ is the threshold, $\mathbf{u}_t = \mathbf{W}_i x_t$ is the input current and $\mathbf{m}_t$ is the membrane potential. We choose the derivative of the $\arctan$ function as the surrogate gradient for $\Theta(\cdot)$ during training. Following Fang et al. (2021a), we have $\Theta'(x) \triangleq \frac{1}{1+(\pi x)^2}$. To improve the expressivity of neurons, we define $\frac{1}{\tau_m} \in (0, 1)^{N_h}$ as a learnable vector, where $\tau_m$ is the membrane time constant.

In our approach, we utilize the step-function characteristic of spiking neurons to generate binary gating signals, while intentionally omitting the reset mechanism. This design choice serves two important purposes. First, the absence of reset operations allows the model to retain historical information, which helps make effective gating decisions across time. Second, this simplification enables the use of efficient parallel scanning algorithms during training, matching the favorable $O(\log T)$ time complexity achieved by linear RNNs while maintaining the benefits of discrete gating.

**Adding Stochasticity.** PSN already meets the desired property of being a gating function. However, due to the absence of stochasticity in its neuronal dynamics, if a neuron outputs 0 at some point during training, it is likely to remain 0 for the rest of the training process. Therefore, we refer to the

trick in discrete model compression Gao et al. (2020), which uses a stochastic threshold to determine the state of discrete gating. We add stochasticity to $V_{th}$, which is now summed by a constant and a stochastic variable during training, and is fixed to a constant during inference based on the expectation of the stochastic variable. Specifically, we have

$$V_{th,i} = \begin{cases} \theta + X_i, & \text{parallel training,} \\ \theta + \mathbb{E}X_i, & \text{sequential inference,} \end{cases} \tag{8}$$

where $V_{th,i}$ is the threshold voltage of the $i$-th neuron, $\theta$ is the threshold potential of the neuron, and $X_i \sim U(0,1)$ is the random variable that has a uniform distribution.

**Linear Recurrent Cell Implementation.** As illustrated in Figure 3 (Right), our linear recurrent cell with discrete gating is implemented by modifying the Simple Recurrent Unit (SRU) Lei et al. (2018). The key modifications are as follows: 1) Replacing continuous gates with discrete gating; 2) Removing non-linear components (*i.e.*, $\mathbf{v}_f$ and $\mathbf{v}_r$) to enable the parallel training; 3) Separating forget gate $c_t$ with input gate $s_t^{in}$. Wrap the memory update (element-wise multiplication $\otimes$ and element-wise addition $\oplus$) with a skip connection controlled by the input gate to enhance memory maintenance. 4) Employing complex-valued hidden states as that in Orvieto et al. (2023); Morad et al. (2023a) and Layer Normalization. $\mathbf{W}_c$, $\mathbf{W}_x$, $\mathbf{W}_i$, $\mathbf{W}_o$ and $\mathbf{W}_y$ represents fully connected layers. Components inside the red box correspond to the spiking neurons defined in Section 4.1, which serve as discrete gates. The triangle symbol denotes a highway connection Srivastava et al. (2015) controlled by the gating function, and LN refers to Layer Normalization. These simplifications and modifications are consistent with our insights on discrete gating. The primary distinction from existing sequence models lies in our use of discrete gating via spiking neurons: a modification aimed at improving memory maintenance by filtering irrelevant information, rather than altering the fundamental memory update mechanism. Further implementation details are provided in Appendix A.

## 5 EXPERIMENTS

Our main experiments are conducted on five memory-intensive RL benchmarks: Passive Visual Match Hung et al. (2019), Repeat Previous, Autoencode, Battleship, and Concentration (the latter four from POPGym Morad et al. (2023b)).

The Passive Visual Match and Repeat Previous tasks require the agent to recall and match a previously presented color or repeat a specific context over extended time horizons. These characteristics make them well-suited for memory maintenance capabilities evaluation, which is the focus of this paper.

In contrast, tasks such as Autoencode, Battleship, and Concentration not only demand long-term memory retention but also require the agent to dynamically update memory states based on complex input streams. These tasks thus help assess the general effectiveness of our method, even with its intentionally simplified recurrent structure for memory updating.

Table 1 summarizes the sequence models included in our main experiments. Among them, FFM Morad et al. (2023a), SHM Le et al. (2025), and LiT Katharopoulos et al. (2020) employ matrix-based memory representations, making them particularly well-suited for handling complex inputs and supporting strong memory update capabilities, a different focus from the scope of our current work.

Complete experimental configurations, including network architectures, reinforcement learning algorithms, and hyperparameters, are provided in Appendix F to ensure reproducibility.

### 5.1 VERIFYING THE PROPOSED SALIENCY-BASED FRAMEWORK

We conducted experiments on Passive Visual Match with a memory length of 250. Training was performed for a total of 1000 episodes, with temporal saliency $R_t$ calculated every 50 episodes. According to Equation 2 in Section 3, we computed both $\mathcal{C}_{in \rightarrow out}$ and $\mathcal{C}_{noise \rightarrow out}$. As shown in Figure 4 (Left), our method achieves the highest $\text{SNR}_\mathcal{C}$ after around 500 episodes, while other methods have lower $\text{SNR}_\mathcal{C}$, reflecting their inability to effectively filter out irrelevant information and, consequently, inferior memory maintenance.

**Non-linear RNNs vs Linear RNNs.** The second and the third plot in Figure 4 (Left) show the values of $\mathcal{C}_{in \rightarrow out}$ and $\mathcal{C}_{noise \rightarrow out}$ during training. It can be observed that the value of $\mathcal{C}_{in \rightarrow out}$ is

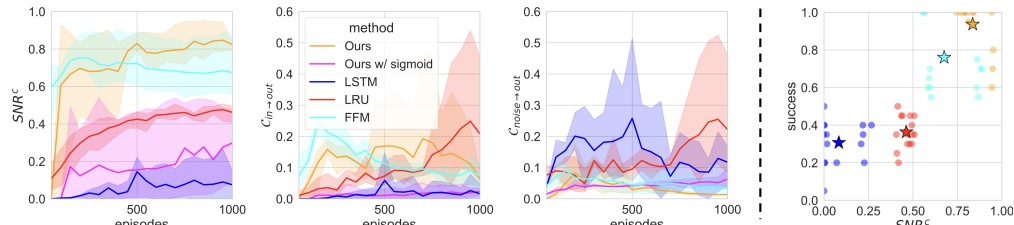

Figure 4: Results of $SNR_{\mathcal{C}}$, $\mathcal{C}_{in \to out}$, and $\mathcal{C}_{noise \to out}$. Correlation between $SNR_{\mathcal{C}}$ and success rate.

very low for LSTM. This is due to the vanishing gradient problem of non-linear RNNs. As a result, $SNR_{\mathcal{C}}$ in Equation 1 of LSTM is the smallest among all methods, indicating that the model can not effectively filter out noise. In contrast, both values are high for LRU, as it alleviates gradient vanishing through linear recurrence. However, since both values remain high for LRU, the $SNR_{\mathcal{C}}$ of LRU is still low. Its ability to filter out noise is still limited.

**Non-Gated RNNs vs Gated RNNs.** We will still focus on the second and the third plot in Figure 4 (Left). For FFM, $\mathcal{C}_{in \to out}$ remains at a similar level compared to LRU, because both of them are linear RNNs. What increases FFM's $SNR_{\mathcal{C}}$ is that $\mathcal{C}_{noise \to out}$ of FFM is much lower than that of LRU, indicating that the gating mechanism helps filter out irrelevant information.

**Continuous Gating vs Discrete Gating.** Nevertheless, due to the continuous nature of the gating in FFM, the value of $\mathcal{C}_{noise \to out}$ fails to reach its optimum. In contrast, the third plot in Figure 4 (Left) shows that our method reduces $\mathcal{C}_{noise \to out}$ to a substantially lower level than FFM, thereby effectively increasing $SNR_{\mathcal{C}}$. This efficient noise filtering ultimately enhances memory maintenance in our approach. To further validate the advantage of discrete gating, we conduct a controlled comparison by replacing the spiking neuron in our model with a sigmoid function, while keeping all other components unchanged. This variant is denoted as **Ours w/ sigmoid**. As shown in Figure 4 (Left), the sigmoid-based gating results in a higher $\mathcal{C}_{noise \to out}$, confirming its ineffectiveness in filtering out noise compared to our discrete gating mechanism.

**Correlation between $SNR_{\mathcal{C}}$ and model performance.** Figure 4 (Right) demonstrates the correlation between $SNR_{\mathcal{C}}$ and method performance. The $SNR_{\mathcal{C}}$-success pair is sampled from the last 5 evaluations of the 3 runs for each method. It can be observed that, given the same number of training steps, a clear positive correlation exists between $SNR_{\mathcal{C}}$ and the success rate. Since our method achieves the highest $SNR_{\mathcal{C}}$, it also achieves the highest success rate under the same training steps, fully demonstrating that using discrete gating to filter out irrelevant information can improve sample efficiency in long-term memory tasks.

## 5.2 COMPARISON UNDER DIFFERENT MEMORY SETTINGS

**Single Long-term Memory Tasks.** In this section, we explore the impact of memory lengths on model performance on Passive Visual Match. In this environment, the first 15 timesteps belong to $P_{in}$ and the last 15 timesteps belong to $P_{out}$, and the size of $P_{noise}$, *i.e.*, memory length, can be set to different values. Figure 5 shows a comparison between our method and other methods under various memory lengths of Passive Visual Match. Specifically, we train each model on tasks with memory lengths of $\{60, 100, 250, 500\}$. It can be observed that when the memory length is short, almost all methods except LSTM exhibit high sample efficiency. However, as the memory length increases, the

| | L | G | M |
|---|---|---|---|
| LSTM Hochreiter & Schmidhuber (1997) | | ✓ | |
| GRU Chung et al. (2014) | | ✓ | |
| LRU Orvieto et al. (2023) | ✓ | | |
| FFM Morad et al. (2023a) | ✓ | ✓ | ✓ |
| SHM Le et al. (2025) | | | ✓ |
| LiT Katharopoulos et al. (2020) | | | ✓ |
| Ours | ✓ | ✓ | |

Table 1: Sequence models used in main experiments. "L" indicates linear recurrence. "G" indicates gating mechanism. "M" indicates matrix-based memory models.

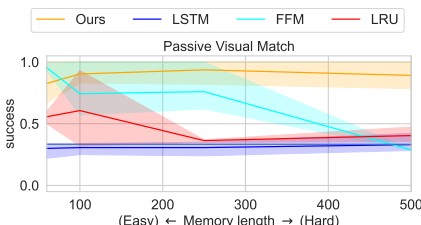

Figure 5: Results of Passive Visual Match with different memory lengths.

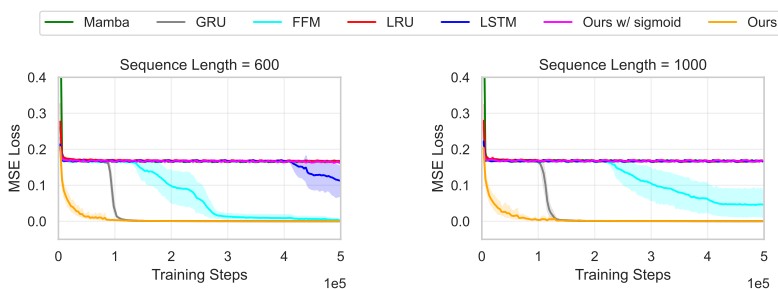

Figure 6: Results of the adding problem with increasing sequence length.

performance of all methods except ours declines significantly. The decline happens first on LRU at a memory length of 250, followed by FFM at a memory length of 500. These results indicate that our method is able to maintain higher sample efficiency than other methods for long-term memory tasks.

Additional experiments on long-term memory tasks include Passive T-maze Ni et al. (2023), a task that is proposed concurrently with Passive Visual Match to test long-term memory and Minecraft Maze Beck et al. (2020), a long-term memory task with more realistic environment (*i.e.*, high-dimensional observation as the input). Those results are provided in Appendix E.1.

**Interleaving Long-term Memory Tasks.** We further conducted experiments to test our proposed method on tasks that have interleaving phases. Following prior works Morad et al. (2023b;a); Le et al. (2025); Morad et al. (2024), we conduct experiments on POPGym Morad et al. (2023b). We refer to the experimental setting of Le et al. (2025) and choose the task "Repeat Previous", where the goal at the $i$-th time step is to output the input from the ($i$-k)-th step. This can be seen as multiple sub-tasks that are stacked together. The memory length of each sub-task is k-1. The task has three levels: easy, medium, and hard, with the distinction being the value of $k \in \{4, 32, 64\}$. A larger $k$ corresponds to a longer memory length and a higher task difficulty. As shown in Table 2, our method achieves strong performance across all three difficulty levels and is the only method that scores above 90 on both the medium and hard tasks. This experimental conclusion aligns with our findings on the Passive Visual Match task, further demonstrating the effectiveness of our method in long-term memory tasks.

**Long-term Tasks with Supervised Learning.** To validate the advantages of our method can generalize to supervised learning tasks that require long-term memory maintenance, we conducted experiments on a classic supervised learning task, the adding problem Hochreiter & Schmidhuber (1997); Bai (2018). This task primarily evaluates the model's ability to retain memory over long sequences with noisy inputs. Its detailed description can be found in Appendix F.2. We adopted the experimental environment and hyperparameter settings provided in Bai (2018), and trained models on different sequence lengths. The compared models include traditional gated RNNs (LSTM Hochreiter & Schmidhuber (1997) and GRU Chung et al. (2014)), linear RNNs (LRU Orvieto et al. (2023)), linear gated RNNs and SSMs (FFM Morad et al. (2023a) and Mamba Gu & Dao (2024)), and the sigmoid-gated variant of our method. Figure 6 illustrates the loss curve of different methods with a sequence length of 600 and 1000. Results of other sequence lengths are provided in Appendix E.3. Our method converges faster than all compared methods on this task, and its relative advantage grows as the sequence length increases. This indicates a significant benefit in satisfying memory maintenance requirements across arbitrary sequence lengths.

**General Memory Tasks.** To verify the general ability of our method in other discrete tasks that require memory. We conduct experiments on tasks (Autoencode, Battleship, Contentration) in POPGym Morad et al. (2023b) following the setting of Le et al. (2025). In these tasks, memory maintenance plays a less critical role compared to long-term memory tasks. Unlike RepeatPrevious, Autoencode requires the agent to reproduce the input sequence in reverse order, a process that not only relies on robust memory maintenance but also on effective memory updating to monitor output progress. Likewise, tasks such as Battleship and Concentration demand enhanced memory update

Table 2: Results of our proposed method compared to existing methods on RepeatPrevious. Methods with ∗ are reported from Le et al. (2025).

| Environment | Level | LiT∗ | GRU∗ | FFM∗ | SHM∗ | Ours |
|---|---|---|---|---|---|---|
| RepeatPrevious | Easy | 6.0±4.0 | **99.9±0.0** | 98.4±0.3 | 88.9±11.1 | 96.2±1.0 |
| | Medium | -46.8±1.1 | -34.7±1.7 | -24.3±0.4 | 48.2±7.2 | **96.6±0.5** |
| | Hard | -48.5±0.3 | -41.7±1.8 | -33.9±1.0 | -19.4±9.9 | **88.3±3.5** |

Table 3: Results of our proposed method compared to existing methods on general memory tasks in POPGym Benchmarks.Methods with ∗ are reported from Le et al. (2025).

| Environment | Level | LiT∗ | GRU∗ | FFM∗ | SHM∗ | Ours |
|---|---|---|---|---|---|---|
| Autoencode | Easy | -44.7±1.4 | -37.9±7.7 | -32.7±0.6 | **49.5±23.3** | 38.7±12.5 |
| | Medium | -47.8±0.2 | -43.6±3.5 | -32.7±0.6 | **-28.8±14.4** | -32.9±1.1 |
| | Hard | -48.1±0.1 | -48.1±0.7 | -47.7±0.5 | **-43.9±0.9** | **-43.9±2.5** |
| Battleship | Easy | -41.3±0.5 | -41.1±1.0 | -34.0±7.1 | **-12.3±2.4** | -35.3±0.5 |
| | Medium | -39.2±0.3 | -39.4±0.5 | -37.1±3.1 | **-16.8±0.6** | -36.9±1.0 |
| | Hard | -38.4±0.2 | -38.5±0.5 | -38.8±0.3 | **-21.2±2.3** | -38.6±0.3 |
| Concentration | Easy | -18.5±0.2 | -10.9±1.0 | **10.7±1.2** | -1.9±2.4 | 3.4±0.9 |
| | Medium | **-18.6±0.2** | -21.4±0.5 | -24.7±0.1 | -21.0±0.8 | -21.3±1.1 |
| | Hard | **-83.0±0.1** | -84.0±0.3 | -87.5±0.5 | -83.3±0.1 | -84.5±0.4 |

capabilities from the agent in order to make accurate decisions. As shown in Table 3, our method performs comparably to SOTA methods. Although SHM achieves better results on Battleship, it is using a matrix-based memory that has a recurrent state size that is 4x larger than ours. To sum up, our method is effective across a wide range of long-term memory tasks.

## 5.3 ABLATION STUDIES

We conducted ablation experiments to evaluate our method, which include: 1) **Discrete Output**: moving the discrete mechanism from gating to the output of the memory model, 2) **Gaussian as SG**: replacing the surrogate gradient function of spiking neurons with the probability density function of the standard normal distribution, and 3) **Constant Vth**: removing the stochastic threshold and replacing it with a constant threshold. The experimental results are shown in Figure 7.

It can be observed that relocating the discrete mechanism from the gating to the RNN output fails to preserve the advantages of discrete gating. This is because the discrete mechanism excels precisely when applied to gating, where it enables more precise control over information selection. If the RNN output is discretized while continuous gating remains in place, filtering is applied to blended information, leading to information degradation rather than improvement.

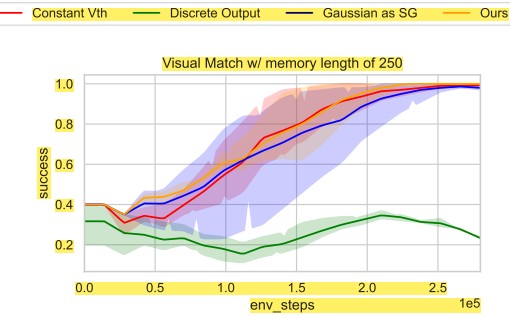

Figure 7: Results of Passive Visual Match by different variants of our method.

It can also be observed that replacing the arctan surrogate function with the Error Function (ERF) results in a slightly lower training curve, although high sample efficiency is still maintained. It is worth noting that our approach does not impose strict constraints on the shape of the surrogate gradient function. These experimental results confirm that our method remains effective across different surrogate choices.

In the experiment where the stochastic threshold is replaced with a constant threshold, the training curve progresses more gradually in the early stages, but eventually rises rapidly in later phases. This suggests that introducing stochasticity can accelerate convergence. However, the core of our method lies in discrete gating itself. Since the discrete nature of the gate does not depend on the threshold mechanism, high sample efficiency is preserved with or without stochasticity, and consistently outperforms methods based on continuous gating.

## 6 DISCUSSION

Long-term memory is an important challenge for partially observable reinforcement learning. In this paper, we analyze why existing sequence models in memory-based RL fail to train efficiently on such tasks and propose spiking neurons as a discrete gating mechanism to solve this problem. Our experimental results underscore the importance of incorporating discrete gating into modern RNN architectures. However, training models with discrete functions is challenging due to the use of surrogate gradient. More limitations are discussed in Appendix G.

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

## A    IMPLEMENTATION DETAILS OF RECURRENT UNITS

We apply highway connections Srivastava et al. (2015) in both temporal and spatial dimensions, which can control two streams with a single gating signal:

$$
\begin{aligned}
h_{t,i} &= \begin{cases} c_{t,i} \odot h_{t-1,i} + (\mathbf{W}_x x_t)_i, & s_{t,i}^{in} = 1, \\ h_{t-1,i}, & s_{t,i}^{in} = 0, \end{cases} \quad h_t \in \mathbb{C}^{N_h}, \\
\hat{y}_{t,i} &= \begin{cases} h_{t,i}, & s_{t,i}^{out} = 1, \\ (\mathbf{W}_x x_t)_i, & s_{t,i}^{out} = 0, \end{cases} \quad \hat{y}_t \in \mathbb{C}^{N_h}.
\end{aligned}
\tag{9}
$$

Here, $i$ refers to the $i$-th element of the cell. $s_t^{in}$ and $s_t^{out}$ refer to the output of the spiking neuron in Section 4.1, their $u_t$ in Equation 7 is calculated as $\mathbf{W}_i x_t$ and $\mathbf{W}_o x_t$, respectively. We follow the setting of existing linear RNNs Orvieto et al. (2023); Morad et al. (2023a) and make our recurrent cell have complex hidden values.

Our method allows multiple ways to construct $c_t$. An approach is to construct an input-independent $c_t$ in the style of LRU Orvieto et al. (2023). We also propose a novel way to construct input-dependent $c_t$, which can be written as

$$
\hat{c}_t = \mathbf{W}_h x_t, \quad \hat{c}_t \in \mathbb{C}^{N_h}, \quad c_t = \hat{c}_t \frac{\tanh(|\hat{c}_t|)}{|\hat{c}_t|}, \quad c_t \in \mathbb{C}^{N_h},
\tag{10}
$$

where $|\cdot|$ is the element-wise absolute value of a complex input. In practice, to avoid the divided-by-zero error, it is implemented as $\sqrt{|\cdot|^2 + 1}$. $\tanh(\cdot)$ is used to clip the elements of $c_t$ inside the unit disk to mitigate value explosion.

Finally, the output $\hat{s}_t$ of our recurrent cell is

$$
\hat{s}_t = \text{LN}(\mathbf{W}_y \cdot \text{Concat}(\text{Re}[\hat{y}_t], \text{Im}[\hat{y}_t])), \quad \hat{s}_t \in \mathbb{R}^{N_{out}},
\tag{11}
$$

where $N_{out}$ is the output size of the sequence model, $\text{Re}[\cdot]$ and $\text{Im}[\cdot]$ are the real and imaginary components of a complex value. LN denotes the nonparametric layer normalization.

It can be proven that the update process in Equation 9 is still parallelizable. The basic idea is to preprocess the input of the associative scan operation, replacing $c_{t,i}$ with 1 and $(\mathbf{W}_x x_t)_i$ with 0 when $s_{t,i}^{in}$ is 0. A detailed proof of associativity can be found in Appendix C.2.

In addition, with the highway connection defined in Equation 9, even if $c_t$ is input-dependent, the conclusion in Proposition 1 still holds. That is, when $s_t$ is a 0 vector, no gradient can propagate from $c_t$ or $(\mathbf{W}_x x_t)_i$ back to $o_t$, allowing $R_t$ to be zero. A detailed discussion is provided in Appendix C.3.

## B PARTIALLY OBSERVABLE MARKOV DECISION PROCESS

Partially Observable Markov Decision Process (POMDP) is a mathematical framework that is used to model the environment that is partially observable by an agent Åström (1965). It formally describes the problem of long-term memory tasks. Since the agent cannot directly perceive the true state of the environment, it must maintain an internal memory of past observations and actions to make optimal decisions. Thus, effectively solving a POMDP inherently requires the ability to perform long-term memory tasks. A POMDP is defined as a tuple $(\mathcal{S}, \mathcal{A}, \mathcal{O}, T, O, R, \gamma)$, where $\mathcal{S}$ represents the state space, which is inaccessible to agents. $\mathcal{A}$ represents the action space. $\mathcal{O}$ represents the observation space of agents. $T = P(s_t|s_{t-1}, a_{t-1})$ is the transition function. $O = P(o_t|s_t)$ is the observation function. $R = P(r_{t+1}|s_t, a_t, s_{t+1})$ is the reward function. $\gamma$ is the discount factor. To obtain an optimal policy, the agent must condition the policy on all accessible information *i.e.*, the trajectory $\tau_t = (o_1, a_1, r_1, \cdots, o_t, a_t, r_t)$ up to timestep $t$. The agent's goal is to learn a policy $\pi(a_t|\tau_t)$ to maximize the expected discounted return, where $T$ is the time horizon.

## C THEORETICAL RESULTS

### C.1 PROOF OF PROPOSITION 1

**Proposition 2.** *If $c_t$ is input-independent, and $\exists \boldsymbol{W}_i \in \mathbb{R}^{N_x \times N_h}$ so that $\sigma(\boldsymbol{W}_i x_t) = 0, t \in P_{noise}$ in a certain environment, $SNR_\mathcal{C}$ can reach its maximum value $1$.*

*Proof.* The expression of $\text{SNR}_\mathcal{C}$ is

$$\text{SNR}_\mathcal{C} = \frac{\mathcal{C}_{in \to out}}{\mathcal{C}_{in \to out} + \mathcal{C}_{noise \to out}}, \mathcal{C}_{in \to out} \geq 0, \mathcal{C}_{noise \to out} \geq 0, \tag{12}$$

According to Equation 2 and Equation 3, $\mathcal{C}_{noise \to out}$ is

$$\mathcal{C}_{noise \to out} = \frac{1}{|P_{noise}|} \sum_{t \in P_{noise}} R_t = \frac{1}{|P_{noise}|} \sum_{t \in P_{noise}} \Big| \sum_{i \in P_{out}} \Big[ \frac{\partial Q(\hat{s}_i, a_i)}{\partial \hat{s}_i} G_t \frac{\partial z_t}{\partial o_t} \Big] \Big| \tag{13}$$

For a memory-based agent with gated linear recurrence defined in Equation 5, which is

$$h_t = c_t \odot h_{t-1} + (\mathbf{W}_x x_t) \odot \sigma(\mathbf{W}_i x_t), \tag{14}$$

the term $G_t$ in Equation 13 is

$$\begin{aligned} G_t =& \frac{\partial \hat{s}_i}{\partial h_i} \Big( \prod_{j=t+1}^{i} \frac{\partial h_j}{\partial h_{j-1}} \Big) \frac{\partial h_t}{\partial z_t} \\ =& \frac{\partial \hat{s}_i}{\partial h_i} \Big( \prod_{j=t+1}^{i} \frac{\partial h_j}{\partial h_{j-1}} \Big) \Big[ \text{diag}\big(\sigma(g_t) \odot \frac{\partial h_t}{\partial u_t}\big) \frac{\partial u_t}{\partial z_t} + \\ & \text{diag}\big(u_t \odot \frac{\partial h_t}{\partial \sigma(g_t)} \odot \sigma'(g_t)\big) \frac{\partial g_t}{\partial z_t} \Big], \end{aligned} \tag{15}$$

where $u_t = \mathbf{W}_x z_t$ and $g_t = \mathbf{W}_i z_t$. $\text{diag}(\mathbf{x})$ denotes the diagonal matrix formed by vector $\mathbf{x}$. Note that $\sigma'(x) = \sigma(x)(1 - \sigma(x))$. Therefore, when $\sigma(\mathbf{W}_i z_t) = 0, t \in P_{noise}$, the red terms $\sigma(g_t)$ and $\sigma'(g_t)$ in Equation 15 are $0$ vectors. We have

$$R_t = 0, t \in P_{noise} \tag{16}$$

$$\mathcal{C}_{noise \to out} = \frac{1}{|P_{noise}|} \sum_{t \in P_{noise}} R_t = 0 \tag{17}$$

$$\text{SNR}_\mathcal{C} = \frac{\mathcal{C}_{in \to out}}{\mathcal{C}_{in \to out}} = 1. \tag{18}$$

$\square$

## C.2 PROOF OF ASSOCIATIVITY OF EQUATION 9

We focus on the first equation in Equation 9, which is:

$$h_{t,i} = \begin{cases} c_{t,i} \odot h_{t-1,i} + (\mathbf{W}_x x_t)_i, & s_{t,i}^{in} = 1, \\ h_{t-1,i}, & s_{t,i}^{in} = 0. \end{cases} \tag{19}$$

It can be rewritten as

$$\begin{aligned} h_t &= (c_t \odot h_{t-1} + \mathbf{W}_x x_t) \odot s_t^{in} + h_{t-1} \odot (1 - s_t^{in}), \\ &= c_t \odot h_{t-1} \odot s_t^{in} + h_{t-1} \odot (1 - s_t^{in}) + \mathbf{W}_x x_t \odot s_t^{in}, \\ &= [c_t \odot s_t^{in} + (1 - s_t^{in})] \odot h_{t-1} + \mathbf{W}_x x_t \odot s_t^{in}, \end{aligned} \tag{20}$$

where $\odot$ is the element-wise multiplication.

Let $\bullet$ be the binary operator that operates on element $e_k$, which is defined as:

$$e_k = (e_{k,a}, e_{k,b}) := \left( c_t \odot s_t^{in} + (1 - s_t^{in}), \mathbf{W}_x x_t \odot s_t^{in} \right). \tag{21}$$

And the operator $\bullet$ is defined as:

$$e_i \bullet e_j = (e_{j,a} \odot e_{i,a}, e_{j,a} \odot e_{i,b} + e_{j,b}). \tag{22}$$

This can be seen as a special case of the scan operator of S5 Smith et al. (2023), where the matrices are all diagonal. Note that $\bullet$ is associative, which means for any element $x, y, z$, we have $(x \bullet y) \bullet z = x \bullet (y \bullet z)$. Thus, the operation can be computed in parallel with a time complexity of $O(\log T)$, where $T$ is the sequence length. The associativity of S5 operator is proven in Smith et al. (2023).

## C.3 DISCUSSION OF DISCRETE GATING

Following Equation 20 and the BPTT algorithm, the term $G_t$ in Equation 3, which relates to the structure of the sequence model is

$$\begin{aligned} G_t &= \frac{\partial \hat{s}_i}{\partial h_i} \Big( \prod_{j=t+1}^{i} \frac{\partial h_j}{\partial h_{j-1}} \Big) \frac{\partial h_t}{\partial z_t} \\ &= \frac{\partial \hat{s}_i}{\partial h_i} \Big( \prod_{j=t+1}^{i} \frac{\partial h_j}{\partial h_{j-1}} \Big) \Big[ \mathrm{diag}\big( s_t^{in} \odot \frac{\partial h_t}{\partial u_t} \big) \frac{\partial u_t}{\partial z_t} + \\ &\quad \mathrm{diag}\big( h_{t-1} \odot s_t^{in} \odot \frac{\partial h_t}{\partial c_t} \big) \frac{\partial c_t}{\partial z_t} \Big], \end{aligned} \tag{23}$$

where $c_t = \mathbf{W}_c z_t$ for an input-dependent $c_t$.

As defined in Ismail et al. (2019), saliency quantifies the contribution of each input to the output, measuring how input perturbations affect model responses. For temporal saliency analysis, we therefore employ the true gradient of the step function (0 almost everywhere, undefined at zero) instead of the surrogate gradient, which is used for training spiking neurons. In this case, the gradient will not propagate through the discrete $s_t^{in}$, and the term $\frac{\partial s_t^{in}}{\partial z_t}$ is omitted.

In this case, when $s_t^{in}$ is a vector filled with 0, we can still have the conclusion of Equation 18. Thus, Proposition 1 still holds for input-dependent $c_t$ with the proposed network structure.

## D COMPARISON OF RELATED WORKS

Table 4 highlights the difference between our method and other related works.

## E ADDITIONAL RESULTS

### E.1 OTHER LONG-TERM MEMORY TASKS

**Passive T-maze.** In this section, we follow Ni et al. (2023) and conduct experiments on the Passive T-Maze. We train all methods for 1000 episodes in an environment with a memory length of 250.

Table 4: Comparison of Related Works. "SL" refers to Supervised Learning and "RL" refers to Reinforcement Learning.

| Method | Learning Method | Discrete Selection Mechanism | Element-wise Gating | Input-dependent Discrete Gating |
|---|---|---|---|---|
| Diet-SNN Rathi & Roy (2020) | SL | | | |
| GLIF Yao et al. (2022) | SL | | ✓ | |
| SkipRNN Campos et al. (2018) | SL | ✓ | | |
| Phased LSTM Neil et al. (2016) | SL | ✓ | ✓ | |
| GRSN Qin et al. (2025) | RL | | ✓ | |
| Ours | RL | ✓ | ✓ | ✓ |

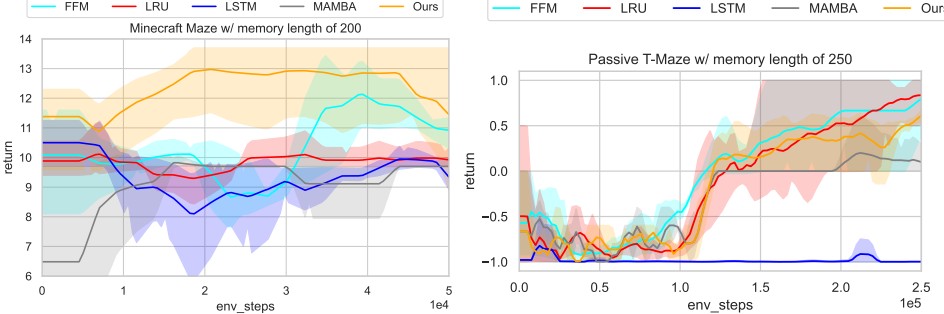

Figure 8: Results of Minecraft Maze Task.     Figure 9: Results of Passive T-maze.

The results are shown in Figure 9. This figure indicates that both our methods and other linear RNNs achieve similar returns within these training steps, whereas the LSTM does not show signs of learning. We attribute this to the fact that the noise phase in the T-Maze provides minimal information, rendering the gating mechanism ineffective in boosting performance for this task. Nonetheless, memory maintenance remains crucial, enabling linear RNNs to outperform LSTMs, as the latter suffers from gradient vanishing issues, causing it to forget information at early time steps.

**Minecraft Maze.** Minecraft Maze Beck et al. (2020) is a set of tasks designed based on Malmo Johnson et al. (2016). It is proposed to evaluate the performance of long-term memory agents in high-dimensional observation spaces and in more realistic environments. At the beginning of each episode ($P_{in}$ in our problem definition), the agent receives an indicator. The agent must then navigate through a series of mazes to obtain short-term rewards ($P_{noise}$ in our problem definition), and finally recall the indicator received at the very start ($P_{out}$ in our problem definition). The maximum memory length required for this task is approximately 200 timesteps.

We compare our method against other methods in the setting of MC-LS in Beck et al. (2020) using the same experimental framework and hyperparameter settings as in Passive Visual Match Ni et al. (2023) with SACD algorithm, detailed in Appendix F. The baselines include LSTM Hochreiter & Schmidhuber (1997), LRU Orvieto et al. (2023), FFM Morad et al. (2023a), and Mamba Gu & Dao (2024). For our method, $\theta$ is set to 0 and the initial value of $\frac{1}{\tau_m}$ is set to 0.1. All methods were trained for 50000 timesteps. As shown in Figure 8, our method achieves not only the highest average return (the peak of the learning curve) but also the highest sample efficiency (the earliest point at which the peak is reached). These results demonstrate the potential of our approach in tackling more complex memory tasks.

### E.2 SHORT-TERM MEMORY TASKS

**Pybullet Tasks.** We verified whether our discrete gating method can perform well on short-term memory tasks. Following Ni et al. (2023); Lu et al. (2024), we conduct experiments in a standard POMDP task used in prior works Ni et al. (2022; 2023); Lu et al. (2024). The hyperparameter settings and other training details are listed in Appendix F.3. We choose Pybullet-P for our verification, where the observation space contains only position information. We compare our method with three methods: LSTM Hochreiter & Schmidhuber (1997), LRU Orvieto et al. (2023), and Transformer Vaswani et al. (2017). The results of LRU and our method are averaged over 3 runs with different

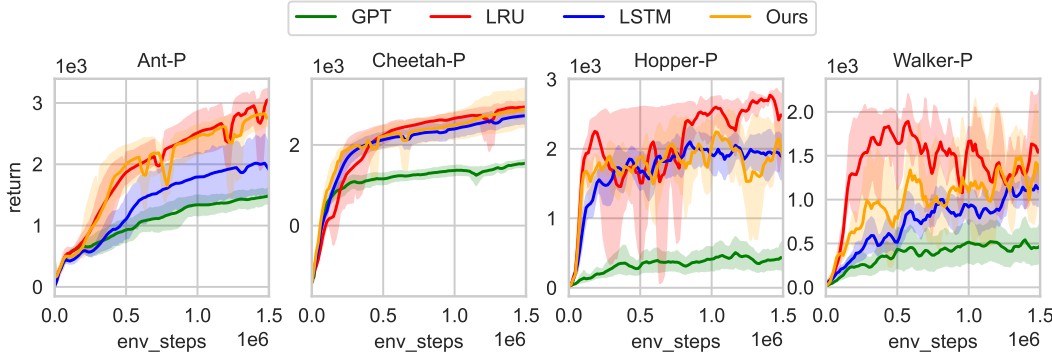

Figure 10: Learning curves of partially observable Pybullet tasks. The shaded area indicates 95% confidence interval.

| Methods | Epochs | Training Time |
|---------|--------|---------------|
| LSTM | 140.7±31.7 | 11.98 |
| GRU | 153.3±45.5 | 16.36 |
| FFM | 164.7±60.3 | 23.85 |
| Mamba | 85.2±28.8 | 131.13 |
| Ours | 120.0±53.6 | 34.50 |

Table 5: Results of 12-AX Task. Training time is measured in seconds.

seeds[1], while the results of LSTM and Transformer are reported from Ni et al. (2023). As shown in Figure 10, our method has similar performance to LRU, a linear RNN without gating, in all environments except Ant-P, but its performance does not greatly surpass that of LRU. This is probably because the inductive bias introduced by discrete gating is more suited for long-term memory tasks that require effective memory maintenance, instead of continual control tasks that require effective memory update. Nevertheless, our method can still outperform LSTM and Transformer in most environments. This validates the design of our linear recurrent cell in Section 4.2.

**The 12-AX Task.** We compare our method with other gated RNNs used in our other experiments (LSTM Hochreiter & Schmidhuber (1997), GRU Chung et al. (2014), FFM Morad et al. (2023a) and Mamba Gu & Dao (2024)) on a classic memory task: 12-AX. This task can be modeled as a supervised learning problem in which the model receives a sequence of symbols drawn from {"1", "2", "3", "A", "B", "C", "X", "Y", "Z"} and must output either "L" or "R" at each time step. After the model receives task signal 1, it must output R at the time step corresponding to X when it encounters the subsequence "AX"; after it receives task signal 2, it must output R at the time step corresponding to Y when it encounters "BY"; at all other times, it should output L. This task tests the model's ability to maintain short-term memory (about 8 timesteps) and to update memory quickly. We followed the experimental setup in O'Reilly & Frank (2006): in each epoch, the model is trained on a sequence with an outer loop of length 25, and training is considered successful when the model's error is zero for two consecutive epochs. We used the Adam optimizer with a learning rate of 1e-3, and set the input and hidden sizes of all RNNs to 64. We use the Mamba implementation provided in SHM Le et al. (2025). Mamba's s6_size is set to 2 to have the same recurrent size as ours. For the unique parameter $V_{th}$ in our method, since this is a supervised learning task that does not require filtering out long-term irrelevant information, we set $V_{th}$ to the constant 0 to improve training efficiency. For each method, we ran 20 trials and recorded the mean number of epochs required for successful training and the total training time.

The results are shown in Table 5. All methods can be trained successfully within a small number of epochs; our method achieves the second-best performance among the gated RNNs, which may be attributable to our simple modeling of memory maintenance. Although Mamba achieves SOTA in the number of epochs, it takes around 4x training time to complete the experiment. In conclusion, our approach effectively balances training efficiency with computational efficiency.

---

[1]We use the JAX implementation provided with Ni et al. (2023).

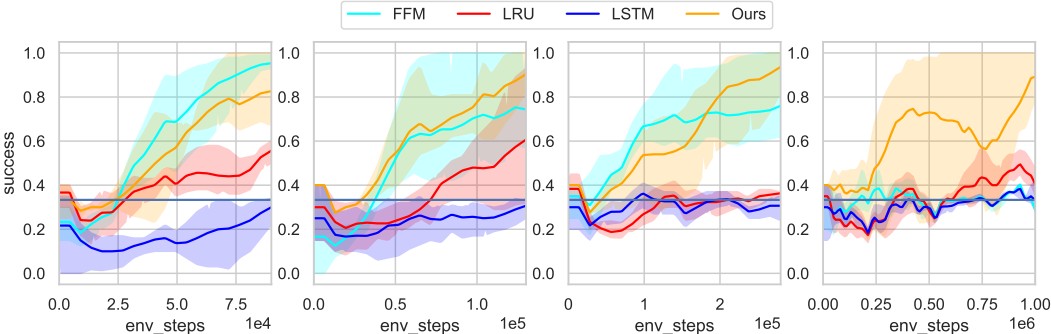

Figure 11: Learning curves of success rate of Passive Visual Match with memory length of 60, 100, 250, and 500. The shaded area indicates 95% confidence interval.

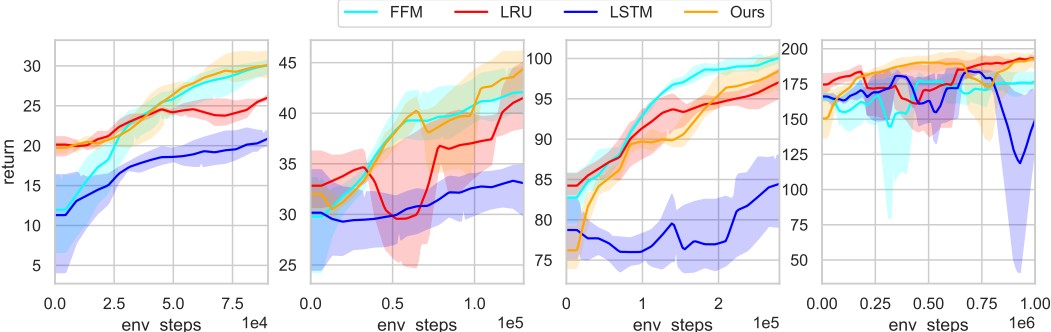

Figure 12: Learning curves of return of Passive Visual Match with memory length of 60, 100, 250, and 500. The shaded area indicates 95% confidence interval.

### E.3 OTHER ADDITIONAL RESULTS

**Learning Curves of Passive Visual Match.** In this section, we show the learning curves of success rate and return of the Passive Visual Match task. For tasks with memory lengths of 60, 100, and 250, the training was performed on a total of 1000 episodes. For the task with a memory length of 500, the training was performed on a total of 2000 episodes. This setting is to make sure the training on each task reaches a stable stage while allowing the evaluation of the final episode to reflect the sample efficiency of each method. The results are shown in Figure 11 and Figure 12. While the success rate has a strong correlation to $SNR_C$, the return does not seem to have that strong correlation. This is because most of the rewards in this environment come from the apple-picking task in Phase 2. This result reflects the sample efficiency of the agent on the memory-independent task. It can be observed that although our method has higher upper bounds on the confidence intervals than the other methods at times during training, the overall difference between our method and the others is marginal, reflecting that the advantage of our method on memory-irrelevant tasks may not be significant.

**The Adding Problem with Other Sequence Lengths.** We present the training loss curves for all sequence lengths used. As illustrated in Figure 13. Our method converges the fastest and maintains an almost constant convergence rate across all sequence lengths, while the convergence rate of other methods degrades markedly as the sequence length increases. This highlights the significant advantage of our method in satisfying memory maintenance requirements for arbitrary sequence lengths.

**Comparison with Mamba.** In this subsection, we aim to evaluate the performance of Mamba Gu & Dao (2024), a class of SSM-based models, on long-term memory tasks. We employed the Mamba model provided in the SHM codebase[2]. To ensure that the size of the recurrent state is roughly consistent with other models, the parameter s6_size was set to 2. We ran this model three times on the Passive Visual Match task under the same experimental settings. As shown in Figure 14, Mamba

---

[2]https://github.com/thaihungle/SHM

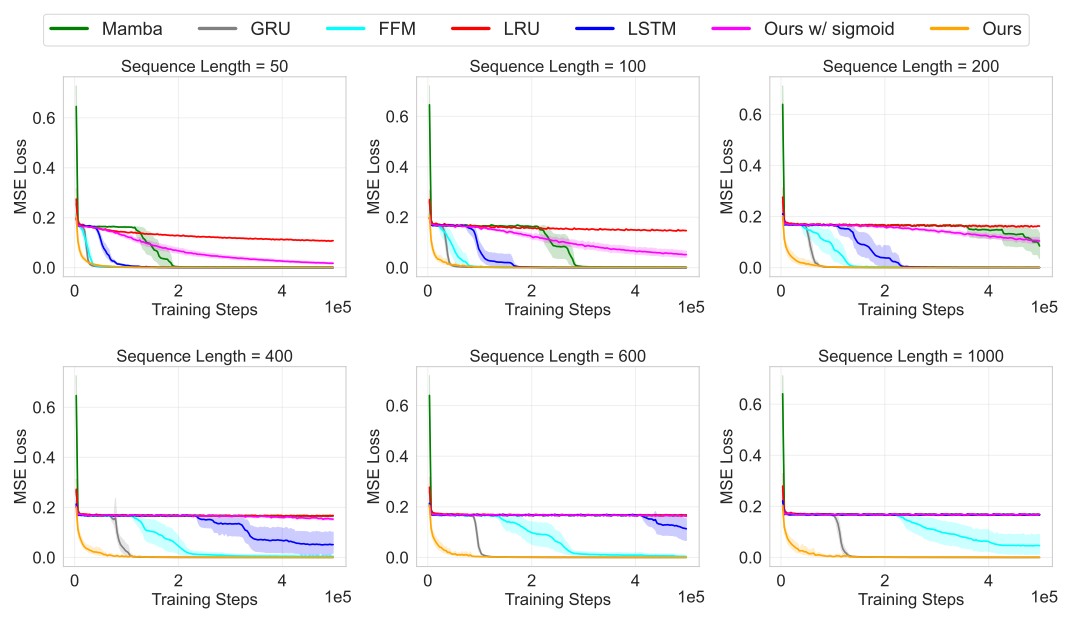

Figure 13: Additional results of the adding problem.

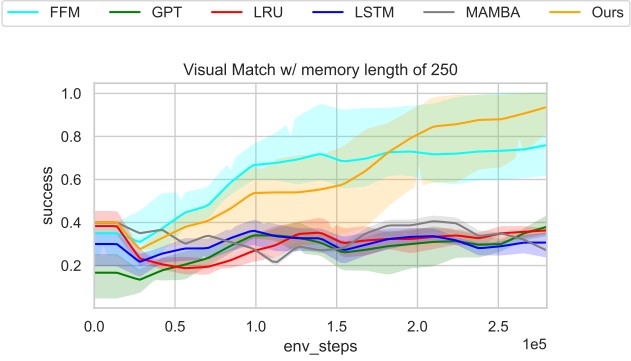

Figure 14: Results on Passive Visual Match including Mamba.

exhibited relatively weak performance on Passive Visual Match and failed to converge within 1000 training episodes. The result is consistent with the results reported in SHM.

Mamba is further evaluated against other methods on the T-Maze, the adding problem, and the 12-AX task, with detailed results presented in Figure 9, Figure 13, and Table 5. In Passive T-Maze, Mamba demonstrates the weakest performance among linear sequence models. In the adding problem, it did not outperform traditional RNNs. In the 12-AX task, it exhibits the slowest training time. Together, these findings highlight the limitations of Mamba in handling certain memory-intensive tasks.

Table 6: Results of computation efficiency on Passive Visual Match.

| Methods | FPS | Memory Consumption |
|---------|-------|--------------------|
| LSTM | 102.74 | 1214 MB |
| LRU | 96.38 | 1910 MB |
| FFM | 125.79 | 1442 MB |
| Ours | 93.2 | 1948 MB |

**Computation Efficiency Analysis.** Like other linear RNNs, our method can leverage parallel scan algorithms to accelerate training, achieving a time complexity of $O(\log T)$ during training and $O(1)$ during inference. The space complexity remains comparable, meaning memory usage grows linearly with the hidden size.

Although the time and space complexity are the same across different linear RNNs, the constant factors in the complexity may vary depending on the implementation of the parallel scan, the number of scan invocations, and the size of the input tensors to the scan operations. Compared to the LRU, which uses a similar PyTorch implementation, our method requires two scan operations per training forward pass—one for the recurrent unit and another for the spiking neurons. However, when keeping the recurrent state size consistent between our method and the LRU, the input tensor size for each scan in our approach is half that of the LRU. Taking these factors into account, the computational constant in our method is slightly larger than that of the LRU, but remains very close.

We recorded the memory consumption and average FPS over the first 200 episodes in Passive Visual Match of length 250. As shown in Table 6, FFM achieves the fastest inference speed. Our method is about 3% slower than LRU and has similar memory consumption as LRU, because our method must perform two parallel scan operations, one for the recurrent unit and one for the parallel spiking unit, while LRU requires only one. Both our method and LRU use PyTorch's implementation of parallel scan. Using a custom CUDA kernel would achieve higher inference speed.

## F EXPERIMENTAL DETAILS

### F.1 DESCRIPTIONS AND SETUPS OF PASSIVE VISUAL MATCH

Passive Visual Match Hung et al. (2019) is divided into 3 phases, the first phase lasts for 15 time steps, where the agent gets a color signal. In the second phase, the agent is transported to a room in which the agent needs to perform the task of picking apples. With each apple picked up, the agent can get a reward of 1. In the third phase, the agent needs to use the color information obtained in the first phase to choose their action. When it reaches the correct location, it receives a reward of 10, and the episode is marked as successful. While the environment is a 7*11 grid world, the agent can only observe a 5*5 space around its location.

Note that the seed for creating the training and evaluation environment in the original implementation[3] is not fixed. This will probably result in different color permutations for training and evaluation. As a consequence, the evaluation result does not correctly reflect the true performance of the agent and has a high variance. In our implementation, we manually set the seed for both training and evaluation environments so that the permutation of the color remains the same. We conducted experiments for all methods on this modified implementation.

### F.2 DESCRIPTIONS OF THE ADDING PROBLEM

In this task, the model input is a sequence of two-dimensional vectors. At each time step, the first element of the vector is a random number between 0 and 1. For the second element, only two time steps in the entire sequence are set to 1 (indicating marked timesteps), while all others are 0. The objective of the task is to remember the value of the first element at the marked timesteps, maintain this memory until the end of the sequence, and finally output the sum of the two remembered values.

### F.3 TRAINING DETAILS

**Hyperparameters.** We use the same RL algorithms and hyperparameter configurations as in prior works Ni et al. (2023); Lu et al. (2024); Le et al. (2025). Specifically, we use SACD Christodoulou (2019) for Passive Visual Match, PPO Schulman et al. (2017) for POPGym and TD3 Fujimoto et al. (2018) for Pybullet. The hyperparameters used in TD3 and SACD algorithms are shown in Table 7. The PPO algorithms' hyperparameters are the same as in Le et al. (2025).

In Table 8, we provide the configuration of the sequence models and network architecture in different environments. For the results that are reported from prior works Ni et al. (2023); Le et al. (2025),

---

[3]https://github.com/twni2016/Memory-RL

Table 7: Hyperparameters of SACD and TD3.

| | Hyperparameter | Value |
|---|---|---|
| | Network Hidden Size | (256, 256) |
| | Batch Size | 64 |
| | Learning Rate | 3e-4 |
| | Replay Buffer Size | 1M |
| | Smoothing Coefficient | 0.05 |
| | Discount Factor | 0.99 |
| SACD | Entropy Temperature | 0.1 |
| TD3 | Exploration Noise | 0.1 |
| | Target Noise | 0.2 |
| | Target Noise Clip | 0.5 |

Table 8: Hyperparameters of sequence models in different tasks. "o", "a" and "r" in input rows refer to the observation, previous action, and reward, respectively. "-" indicates the model is not used in this experiment, or the value is not in the model configuration.

| Environment | Hyperparameter | LSTM | GRU | LRU | FFM | GPT | LiT | SHM | Ours |
|---|---|---|---|---|---|---|---|---|---|
| Passive Visual Match | Inputs | | | | o | | | | |
| | Embedding Size | | | | 100 | | | | |
| | Sequence Length | | | | 90, 130, 280, 530 | | | | |
| | Num Layers | | | | 1 | | | | |
| | Hidden Size | 256 | - | 200 | 128 | | - | - | 128 |
| | Num Heads | - | - | - | - | - | - | - | - |
| POPGym | Inputs | | | | o | | | | |
| | Input Size | | | | 128 | | | | |
| | Sequence Length | | | | 1024 | | | | |
| | Num Layers | | | | 1 | | | | |
| | Recurrent State Size | - | 256 | - | 1024 | - | 256 | 16384 | 4096 |
| Pybullet | Inputs | oa | - | oar | - | oa | - | - | oar |
| | Sequence Length | | | | 64 | | | | |
| | Embedding Size | [32, 16] | - | [64, 16, 16] | - | [64, 64] | - | - | [64, 16, 16] |
| | Hidden Size | 128 | - | 256 | - | 128 | - | - | 128 |
| | Num Layers | 1 | - | 2 | - | 1 | - | - | 2 |
| | Num Heads | - | - | - | - | 1 | - | - | - |

*e.g.*, LSTM and GPT in Pybullet tasks, we copy their configurations from the original paper Ni et al. (2023).

In Passive Visual Match, the implementation of LRU is directly extracted from the code of Lu et al. (2024)[4], which forces the hidden size to be the size of the embedding size multiplied by 2. We keep this setting so the hidden size of LRU is 200 in this environment.

The "Recurrent State Size" in Table 8 refers to the size of the recurrent state of an RNN-like sequence model, flattened to a vector and converted to the **float32** data type. SHM Le et al. (2025) is a matrix-based memory module. The matrix size of SHM used in POPGym tasks is 128*128, indicating that the state size is 16384. To calculate the state size of our module, we treated the membrane potential of input and output gates as a part of the hidden state, so the "actual hidden size" (similar to the definition of LSTM and GRU) of our module for POPGym tasks is 1024. Although it is still larger than that of GRU and FFM, the hyperparameter tuning experiment in SHM Le et al. (2025) showed that their performance will not increase much as their state size increases.

One unique hyperparameter has been introduced in our proposed method, namely, $\theta$, which controls the threshold of the spiking neuron. In Passive Visual Match tasks, $\theta$ is set to 1, resulting in an expected threshold of 1.5. In other environments, $\theta$ is set to 0, resulting in an expected threshold of 0.5. The $\theta$ is set higher in Passive Visual Match since we want to make spiking neurons have

---

[4]https://github.com/CTP314/TFPORL

sparser output, the hidden state will then be changed less frequently, and the model should have better long-term dependency. For the supervised learning experiments: the adding problem and the 12-AX task, we find it preferable to keep $V_{th}$ fixed. Specifically, $V_{th}$ is set to 0.1 for the adding problem and 0.5 for the 12-AX task.

Another hyperparameter of our method is the initial value of $\frac{1}{\tau_m}$, which controls the speed of change of the gating outputs at the beginning of the training. In most of the experiments, this value is set to 0.5. In the adding problem and the Minecraft Maze task, this value is tuned separately and is set to 0.1.

**Compute Resource.**  We run all our experiments on NVIDIA 3090 and 5090 GPUs. For the Passive Visual Match task with a memory length of 250, a single run of our method (1000 episodes) on a single 5090 GPU would take approximately 1 hour. For the POPGym task, when running 2 experiments in parallel on a single 3090 GPU, both experiments would be completed in 7 hours. For the Pybullet task, when running 4 experiments in parallel on a single 3090 GPU, these experiments can be completed in approximately 2 days.

Passive Visual Match and Pybullet require less GPU memory and could theoretically run more experiments in parallel, but a roughly linear increase in total run time is also observed. POPGym, on the other hand, requires more GPU memory, and increasing the number of parallelizations may result in CUDA out-of-memory errors.

# G    LIMITATIONS

While the experiments presented in the paper confirm the performance of the proposed method, there are also some limitations.

**Efficient Memory Update.**  Although our approach demonstrates strong memory maintenance capabilities in certain long-term memory tasks, it often falls short of achieving optimal performance in general memory tasks and short-term memory tasks, which require efficient memory updating. In the future, we will aim to enhance the model architecture to improve its memory update ability across a broader range of memory tasks.

**Training Instability.**  Discrete gating and surrogate gradient could potentially introduce instability during training. It also causes the divergence of the Q value in some runs of continual control tasks. Those invalid runs are discarded in our experimental results. Stabilizing the training process is important to enhance the practical use of discrete gating.

**Long-term Temporal Credit Assignment.**  In early exploratory experiments, we found that our approach did not have similar performance gains over existing methods in long-term temporal credit assignment tasks Ni et al. (2023), such as Key-to-Door Raposo et al. (2021). Future work could attempt to combine the proposed module with algorithms specialized for temporal credit assignment to address this challenge.

# H    USE OF LLMS

The use of LLM in this paper (first version) was limited to correcting grammar, polishing the text, and translating some sections from drafts written by the authors into English.

During the rebuttal period, we use LLMs to generate a coarse experimental framework of the 12-AX task, and some bugs are fixed by the authors.

