# OpenReview forum: "Spiking Neuron as Discrete Gating for Long-Term Memory Tasks"
_ICLR.cc/2026/Conference — Submitted to ICLR 2026_

### Official Review · Reviewer_oAXs · 2025-10-28

**Soundness:** 3
**Presentation:** 3
**Contribution:** 3
**Rating:** 4
**Confidence:** 4

**Summary:**

This study focuses on gated neural memory models, noting that the nature of the gate of such models strongly influences the effective time-duration over which a model can learn. Specifically, with sigmoidal gates, true "gating" is nearly impossible and requires very large gating-weights, which hinders learning. Instead, the authors propose to use spiking neuron models combined with surrogate gradient learning to learn discrete gating in memory neuron models. The effectiveness of this approach is demonstrated on the Passive Visual Match (PVM) task and the POPGym. The approach excels on the PVM task compared to other gated networks, especially for longer durations, and also on the RepeatPrevious memory task.

**Strengths:**

The notion of learning discrete gating using surrogate gradients is novel and promising as far as I know. In the shown example, the presented approach excels exactly where the problem of noisy interference in a task is clear.

**Weaknesses:**

The writing would benefit from a more concise writeup of the introduction and clarity of the aims, as well as a more elaborate writeup of the tasks and results. This would also allow some of the other results, like short-term memory, to be included in the main text.

In particular, while the approach does indeed work well for the specific scenario it was designed for, inputs followed by noisy delays followed by a relevant output phase, the performance on general memory tasks is not convincing compared to the noted SHM approach in Table 3, while the presented method seems to perform similar to GRU, LiT and FFM for harder versions of the tasks . It would be informative to determine what makes SHM better.

Gated-memory networks cover a large part of deep learning with many other tasks besides those shown. I would have like to see other memory tasks as well, for example  like the 1-2AX, saccade-anti-saccade tasks, the original "add" task.

**Questions:**

Figure 1 seems to suggest that the approach is mostly focused on a very specific trial type: a signal, noise, and then a go signal/output phase. Does the approach generalize?

In the description of the spiking neuron, I don't see a reset. Is there a reset? If not, what is the effect?

Can you provide more insight into when the gating helps and when not, compared to other methods?

How does the proposed method perform on other long-term memory tasks?

---

> ### Author Response · Authors · 2025-11-21
>
> We thank the reviewers for their thoughtful feedback on our manuscript. The comments have been addressed with corresponding revisions, which are highlighted in the updated PDF.
>
> Our detailed responses to each point are provided below.
>
> ---
> ### W1: The writing would benefit from a more concise writeup of the introduction and clarity of the aims, as well as a more elaborate writeup of the tasks and results.
>
> - Thank you for the advice. We made the following modifications to our paper in the revised version:
>
> **Revision of W1 (summarized in Summary 3.3)**
> 1. We clarify the aim of our paper: solving the problem of efficient memory maintenance in RNN-based models for long-term memory RL tasks.
>      > Revision in paper: line 30-39, Section 1
> 2. We use simpler sentences to describe the results of the experiments. Each result is followed by a brief analysis that links the result to our theoretical findings.
>      > Revision in paper: line 391-392, 395-396, 405-406, Section 5.1
>
> ---
> ### W2: The result of general memory tasks are not convincing. It would be informative to determine what makes SHM better.
>
> - In those general cases, a single timestep may contain both useful information and noise. The agent may need to effectively update memory for complex decision-making. Memory updating is not the problem we are trying to solve in this paper. Despite this, our method achieves performance close to FFM, the state-of-the-art linear gated RNN.
>
> - SHM has a larger memory capacity (a 128×128 memory matrix), allowing more memories to be stored without pre-filtering irrelevant information through gating. In addition, SHM incorporates several designs to prevent gradient instability, such as clamping the elements in the calibration matrix within [0, 2] and using stochastic parameters to reduce dependency between timesteps. These improvements give SHM stronger memory update capabilities. Those contributions are orthogonal to ours.
>
> **Revision of W2 (summarized in Summary 3.1 and 3.2)**
> 1. To better explain the results of the general memory tasks, we revise the description of Autoencode, Battleship, and Concentration, explaining why they require effective memory updating.
>      > Revision in paper: line 361-364, 472-475, Section 5
> 2. We revised the statements analyzing SHM's memory capacity, emphasizing the benefits of larger capacity for memory updating.
>      > Revision in paper: line 362-365, 477, Section 5
>
> ---
> ### W3: Other memory tasks should be added (e.g., 12-AX, saccade-anti-saccade tasks, the original "add" task).
>
> - Thank you for the suggestion. These tasks are classic memory tasks. According to the experimental setups described in [1, 2, 3], 12-AX and the adding problem are Supervised Learning tasks. The memory length required for the 12-AX and saccade-anti-saccade tasks is around 10, so we classify them as short-term memory tasks.
>
> - Although solving these tasks were not the original motivation for our method, we are curious about how our approach performs on them. So far, we have conducted experiments on the 12-AX task, comparing our method with other gated RNNs. We followed the experimental setting in [1]. The results show that our approach effectively balances training efficiency with computational efficiency. If you have any suggestions for additional tasks (preferably those that are open-sourced), we would greatly appreciate your input.
>
> **Revision of W3 (summarized in Summary 4.3)**
> 1. We provide a detailed report of the 12-AX task.
>      > Revision in paper: line 964-970, 994-1011, Appendix E.4
>
> [1] O'Reilly, Randall C., and Michael J. Frank. "Making working memory work: a computational model of learning in the prefrontal cortex and basal ganglia." Neural computation 18.2 (2006): 283-328.
>
> [2] Rombouts, Jaldert, Pieter Roelfsema, and Sander Bohte. "Neurally plausible reinforcement learning of working memory tasks." Advances in neural information processing systems 25 (2012).
>
> [3] Bai, Shaojie, J. Zico Kolter, and Vladlen Koltun. "An empirical evaluation of generic convolutional and recurrent networks for sequence modeling." arXiv preprint arXiv:1803.01271 (2018).

---

> > ### Author Response · Authors · 2025-11-21
> >
> > ### Q1: Figure 1 seems to suggest that the approach is mostly focused on a very specific trial type: a signal, noise, and then a go signal/output phase. Does the approach generalize?
> >
> > - We indeed conducted our analysis under a specialized task setting. This is because our focus is on the problem of memory maintenance, and we simplified other aspects of memory (such as memory updating) to better model this issue.
> >
> > - Although the task setting we propose is specialized, the problem we study—memory maintenance—is universally present in long-term memory tasks. Intuitively, the longer the temporal span of a task, the more important memory maintenance becomes compared to memory updating. Stronger memory maintenance ability translates into greater practical utility in real-world environments.
> >
> > - In the future, we plan to enhance other aspects of memory in our method, such as memory updating.
> >
> > **Revision of Q1 (summarized in Summary 1.1)**
> > 1. We highlight the importance of modeling effective memory maintenance.
> >      > Revision in paper: line 31-37, Section 1
> > 2. We describe how we simplify the problem by assuming crucial information or memory can be updated readily.
> >      > Revision in paper: line 68-69, Caption of Figure 1
> >
> > ---
> > ### Q2: In the description of the spiking neuron, I don't see a reset. Is there a reset? If not, what is the effect?
> >
> > In the spiking neurons we used, reset is indeed removed. This design choice was made for two reasons:
> >
> > - To enable parallel scan and match the time complexity of linear RNNs during training.
> >
> > - To retain as much historical information as possible for gating.
> >
> > **Revision of Q2 (summarized in Summary 5.1)**
> > 1. We highlight that this is a spiking neuron model without reset.
> >      > Revision in paper: line 319-320, Section 4.2
> > 2. We attach the above reasons to explain the removal of reset.
> >      > Revision in paper: line 320-323, Section 4.2
> >
> > ---
> > ### Q3: Can you provide more insight into when the gating helps and when not, compared to other methods?
> >
> > We appreciate the opportunity to summarize the strengths and limitations of our method. The gating in our approach is discrete gating, specifically designed for the problem of memory maintenance.
> >
> > - Advantages of discrete gating: Compared to non-gated methods, gating can more effectively filter out irrelevant information. Compared to continuous gating, our method can filter irrelevant information more efficiently. Therefore, in environments with long noise phases, our method performs particularly well.
> >
> > - Limitations of discrete gating: Unlike continuous gating, our method requires surrogate gradient training, which inevitably introduces approximation errors. This may hinder the model from learning the correct memory update patterns. Consequently, if a task requires integrating information across multiple timesteps for decision-making, our method may not demonstrate its full advantage.
> >
> > **Revision of Q3 (summarized in Summary 1.1 and 3.4)**
> > 1. We highlight the design goal and the task setting.
> >      > Revision in paper: line 31-37, Section 1 and line 68-69, Caption of Figure 1
> > 2. We Add limitations discussion of efficient memory update.
> >      > Revision in paper: line 1154-1158, Appendix G
> >
> > ---
> > ### Q4: How does the proposed method perform on other long-term memory tasks?
> >
> > - Our focus is on memory maintenance rather than memory updating. Although there are many long-term memory tasks in the field, we selected those most aligned with the problem we aim to solve.
> >
> > - Memory maintenance is universally present in long-term memory tasks. Stronger memory maintenance ability translates into greater practical utility in real-world environments. Moreover, memory maintenance and memory updating exhibit a certain degree of independence, so they can be studied separately.
> >
> > - We adopt the simplest design for memory updating. The recurrent unit structure is only minimally modified from the Simple Recurrent Unit (SRU) to support parallel training and memory maintenance.
> >
> > - Despite this simple design, our method remains highly competitive in experimental results.
> >
> > - In addition to the two main tasks, we conducted the Passive T-Maze experiment. The results show that our method is competitive against all linear RNN-based methods. If there are suggestions regarding the placement of this experiment, we will adjust it in subsequent revisions.
> >
> > **Revision of Q4 (summarized in Summary 1.2 and 4.3)**
> > 1. We emphasize the simplicity of our recurrent unit design.
> >      > Revision in paper: line 337-350, Section 4.2
> > 2. We include the Passive T-Maze experiment.
> >      > Revision in paper: line 899-904, 906-950, Appendix E.2

---

### Official Review · Reviewer_ELid · 2025-11-01

**Soundness:** 3
**Presentation:** 3
**Contribution:** 3
**Rating:** 8
**Confidence:** 4

**Summary:**

In this work, the authors suggest that noisy distractor observations greatly reduce the efficiency of memory models. To mitigate this issue, they propose using memory with discrete (spiking) neurons.

First, the authors introduce a signal-to-noise analysis framework for memory. It uses a long-term task with known credit assignment, and is relatively straightforward. It takes the gradient with respect to the observations of the correct credit assignment divided by the total credit assignment.

Next, they perform a theoretical analysis of SNR for linear and nonlinear RNNs, and then gated and non-gated RNNs. They focus on the effect of SNR and the gradient for both cases. Importantly, they prove that gating enables an SNR of 1, which is not possible with a non-gated recurrent update.

The authors go on to motivate a **discrete** and associative gating mechanism called the parallel spiking neuron. It uses a heaviside step gating mechanism surrogate gradient to enable backpropagation. They add stochasticity to mitigate an issue where a neuron can be stuck at 0, preventing gradient flow back through the RNN. They use this mechanism to construct a linear recurrent cell.

The authors calculate the SNR for their model with both soft and hard gating, as well as other prior work. Their method obtains greater SNR than other methods. Nonlinear RNNs demonstrate vanishing gradients, and gated linear RNNs report better SNR than nongated linear RNNs. Finally, discrete gating produces slightly better SNR than sigmoidal gating.

They report returns for other tasks, demonstrating their cell can learn more quickly than other models as the temporal dependence length increases. They perform further comparisons on POPGym tasks.

**Strengths:**

This paper is well written, theoretically sound, and novel. In particular, I find viewing RNN performance through the lens of SNR interesting, and it serves as good motivation for the proposed RNN cell. Table 3 is also particularly refreshing, demonstrating that the proposed method is designed for high-noise scenarios and does not necessarily achieve SoTA performance on standard benchmarks. Finally, the authors provide code for reproducibility.

**Weaknesses:**

More experiments are always beneficial. I think exploring other surrogate gradients would be interesting (why $\arctan$ instead of $\tanh$ or $\mathrm{erf}$?) I think an ablation for introducing noise would also be useful. I believe Stable Hadamard Memory [1] already performs such an ablation, but it would be interesting to see how important it is for discrete gating to work.

## References
[1] Hung et al., Stable Hadamard Memory: Revitalizing Memory-Augmented Agents for Reinforcement Learning

**Questions:**

I think figure 1 has a caption that is far too long.

---

> ### Author Response · Authors · 2025-11-21
>
> We sincerely appreciate the reviewers' time and constructive feedback on our manuscript. We have carefully organized the comments, responded to each, and implemented the necessary revisions. These changes are highlighted in the revised manuscript.
>
> Our specific responses are outlined below:
>
> ---
> ### W1: More experiments are always beneficial (ablation study on surrogate gradient and introducing noise).
>
> Thank you for the suggestions. We add two more ablation studies on the Passive Visual Match task, here is the observation:
>
> - Using error function (ERF) as surrogate function will have similar results. This might indicate that any bell-shaped function as surrogate function is valid.
>
> - Removing the stochasticity in threshold have a little impact on the early stage of training, but the learning curve and the sample efficiency are very close to the original method.
>
> **Revision of W1 (summarized in Summary 4.2):**
> 1. We provide the results and detailed discussions.
>     > Revision in paper: line 505-515, Section 5.3
>
> ---
> ### Q1: Figure 1 has a caption that is far too long.
>
> - Thank you for pointing out this issue. We agree that layout have a significant impact on the overall presentation of the article.
>
> **Revision of Q1 (summarized in Summary 2.2):**
> 1. We split Figure 1 into two figures. The left part is still in Figure 1, but the right part will be moved to Section 3, since it is used to illustrate the problem of different RNNs that we analysed in that section.
>     > Revision in paper: line 162-175 on Page 4

---

> > ### Comment · Reviewer_ELid · 2025-11-24
> >
> > Thank you for the additional ablations. I think this is a good paper and I maintain my score of accept.

---

> ### Author Response · Authors · 2025-11-24
> **Official Comment by Authors**
>
> We sincerely appreciate your continued support and positive feedback. Your timely feedback on our response means a great deal to us, and we are very grateful for your encouragement.
>
> If any new questions arise after you see the other reviewers' comments (or coming comments), we would be pleased to provide clarifications or conduct any additional experiments.

---

### Official Review · Reviewer_gBTK · 2025-11-01

**Soundness:** 2
**Presentation:** 2
**Contribution:** 1
**Rating:** 2
**Confidence:** 4

**Summary:**

The paper introduces a linear recurrent module that uses Spiking Neurons as discrete, input-dependent gating mechanisms. On the Passive Visual Match task with varying memory lengths, showing superior performance as length increases. Ablation studies demonstrating that their discrete gate outperforms a continuous (sigmoid) version. Based on the existing limitations and issues discussed below, I recommend reject.

**Strengths:**

* The work is the first to explore Spiking Neurons as a discrete gating mechanism within a linear RNN framework for memory-based reinforcement learning. This is a interesting architectural choice.
* It achieves excellent performance on specific long-term memory tasks, such as RepeatPrevious task in POPGym.

**Weaknesses:**

* It lacks comparison with existing state-of-the-art linear-time sequence models, such as Mamba2.
* It is missing an quantitative analysis of computational efficiency.
* It achieves SOTA performance only in a minority of environments; its performance in other environments is inferior to baselines, which limits the general applicability of the method.

**Questions:**

* The motivation for using Spiking Neurons is unclear. Combining a simple RNN or SSM with a discrete output could also achieve the effect of discrete gating. Why use Spiking Neurons, which may introduce additional training instability?
* Why does the method perform poorly in other tasks? Although the authors claim comparable performance to LRU on MuJoCo tasks, the implementation of LRU is much simpler and its performance is superior.
* Mamba's selection mechanism also performs information filtering. Does the authors' method filter noise more effectively than Mamba?
* The method performs poorly on tasks like 'Autoencode' and 'Battleship', underperforming SHM. Does this indicate that the method is only effective for tasks with explicitly defined noise phases?
* The paper mentions in Appendix E that training instability and Q-value divergence occurred in some runs, leading to the discarding of those results. Selectively discarding "invalid" runs, rather than reporting results from all random seeds, may overestimate the average performance of the method and makes the comparison with baseline methods unfair.

---

> ### Author Response · Authors · 2025-11-21
>
> We are grateful to the reviewers for carefully reading our manuscript and offering valuable feedback. We have compiled the comments, provided detailed responses, and made the corresponding revisions. All changes are highlighted in the updated PDF.
>
> Our point‑by‑point responses are presented below.
>
> ---
>
> ### W1: It lacks comparison with existing state-of-the-art linear-time sequence models, such as Mamba2.
>
> - Mamba2 is not SOTA in the RL framework we used (Recurrent Model-Free RL). To the best of our knowledge, there is no research on RNN-based memory RL using Mamba2. If such experiments are required, we can implement a model based on the Mamba2 architecture in [1] and update the results accordingly.
>
> - Existing SSMs are not SOTA in the Recurrent Model-Free RL as well (though there have been studies of Mamba in other related fields[2, 3, 4]). Prior work (SHM) compared Mamba against FFM and other methods, showing that Mamba did not outperform FFM in Passive Visual Match. We therefore selected FFM, which demonstrated superior performance in our setting. Our quick verification using the Mamba implementation from SHM confirmed that, with comparable memory size, Mamba underperforms on Passive Visual Match. Additional experiments on T-Maze yielded consistent results.
>
> **Revision of W1 (summarized in Summary 4.1):**
> 1. We provide analysis, discussion, and comparison about Mamba2.
>     > Revision in paper: line 97-98, line 113-117, Section 2
> 2. We provide results and related discussion.
>     > Revision in paper: line 1012-1024, Appendix E.5
>
> [1] Dao, Tri, and Albert Gu. Transformers are ssms: Generalized models and efficient algorithms through structured state space duality. arXiv:2405.21060 (2024).
>
> [2] Huang, Sili, et al. Decision mamba: Reinforcement learning via hybrid selective sequence modeling. NeurIPS 37 (2024): 72688–72709.
>
> [3] Dai, Yang, et al. "Is mamba compatible with trajectory optimization in offline reinforcement learning?." Advances in Neural Information Processing Systems 37 (2024): 51474-51502.
>
> [4] Wang, Wenlong, et al. "Drama: Mamba-enabled model-based reinforcement learning is sample and parameter efficient." arXiv preprint arXiv:2410.08893 (2024).
>
> ---
> ### W2: Missing quantitative analysis of computational efficiency.
>
> - As a parallel scan-based method, our approach has the same time complexity as other linear RNNs.
>
> - We provide constant-time analysis (see response to Reviewer M6hE Q1).
>
> - We report average FPS over the first 200 episodes on Passive Visual Match with memory length 250. Our inference speed is ~3% slower than LRU.
>
> **Revision of W2 (summarized in Summary 4.4):**
> 1. Add complexity and constant analysis.
>     > Revision in paper: line 1026-1053, Appendix E.6
> 2. Provide experimental results.
>     > Revision in paper: Table 6 on Page 20
>
> ---
> ### W3: It achieves SOTA performance only in a minority of environments.
>
> - Our focus is not on general memory tasks but on memory maintenance, i.e., filtering irrelevant information. General and short-term tasks require efficient memory updating, which is outside our scope.
>
> - While results on general tasks are limited, our contributions extend beyond experiments. We provide theoretical insights showing why discrete gating mechanisms are more effective for noise filtering, enabling long-term memory maintenance. The observed results align with our theoretical analysis and expectations.
>
> **Revision of W3 (summarized in Summary 1.1):**
> 1. Highlight our problem focus in Section 1.
>     > Revision in paper: line 30-39, Section 1
> 2. Rephrase contributions in Introduction.
>     > Revision in paper: line 175-180, Section 1

---

> > ### Author Response · Authors · 2025-11-21
> >
> > ### Q1: The motivation for using Spiking Neurons is unclear.
> >
> > - Our theoretical analysis (the memory maintenance analysis through the lens of gradient backpropagation) shows discreteness should be applied to gating, not other components, such as a output of the RNN. Discrete gating fully discards irrelevant information, whereas continuous gating mixes noise and useful signals.
> >
> > - To further substantiate this point, we conducted an ablation study where we moved the discreteness from the gating mechanism to the model’s output. It confirms the necessity of discrete gating.
> >
> > **Revision of Q1 (summarized in Summary 1.2):**
> > 1. Emphasize why our method suits memory maintenance in Section 4.
> >     > Revision in paper: line 292-294, line 300-301, Section 4.1
> > 2. Provide ablation study results (Discrete Output) in Appendix C.6.
> >     > Revision in paper: line 494-503, Section 5.3
> >
> > ---
> > ### Q2: Why does the method perform poorly in other tasks?
> >
> > - Our method focuses on memory maintenance, performing best on long-term tasks with such demand.
> >
> > - Short-term tasks demand efficient memory updating. Methods with strong memory update strategies (e.g., LRU with careful initialization) perform better in such cases.
> >
> > - Our contribution lies in theory and performance for memory maintenance tasks. The memory maintenance analysis framework proposed in this paper is well-developed and advanced.
> >
> > **Revision of Q2 (summarized in Summary 1.1 and 3.1):**
> > 1. Highlight our focus in the Introduction.
> >     > Revision in paper: line 30-39, Section 1
> > 2. Add discussion of short-term tasks.
> >     > Revision in paper: line 482-484, Section 5.2 and line 959-961, Appendix E.3
> >
> > ---
> > ### Q3: Mamba's selection mechanism also performs information filtering. Does the authors' method filter noise more effectively than Mamba?
> >
> > - Our method focuses on memory maintenance, which can be analyzed under the framework of gradient backpropagation proposed in Section 3.1.
> >
> > - Mamba’s selection mechanism is continuous. If a selction mechanism is implemented with continuous function, the problem illustrated in Figure 3 will occur. That is the gradient vanishing problem happened when the continuous function trys to completely filter out irrelevant information.
> >
> > - Our analysis shows that discrete gating avoids this issue, maintaining stronger gradient signals. As eviences in Experiment results of Section 5.1.
> >
> > - We conduct additional comparison with Mamba, The results confirm our method outperforms Mamba in Passive Visual Match and T-Maze regarding sample efficiency.
> >
> > **Revision of Q3 (summarized in Summary 4.1 and 5.2):**
> > 1. We add discussion of Mamba.
> >     > Revision in paper: line 1012-1024, Section E.5
> > 2. We provide results of Mamba.
> >     > Revision in paper: Figure 10 on Page 18, Table 5 and Figure 11 on Page 19
> > 3. We revised our analysis on gradient, linking it more closely to memory maintenance.
> >     > Revision in paper: line 211-215, Section 3.1 and line 292-294, 300-301, Section 4.1
> >
> > ---
> > ### Q4: Does this indicate that the method is only effective for tasks with explicitly defined noise phases?
> >
> > - That is true based on our experiment results.
> >
> > - The reason is that our research focus lies in the issue of memory maintenance. By modeling such problems as memory tasks, we simplify other aspects of memory (such as memory updating).
> >
> > - Although the task setting we propose is specialized, the problem of memory maintenance exists universally across various long-term memory tasks.
> >
> > - Intuitively, the longer the time span of a task, the greater the importance of memory maintenance compared to memory updating. Stronger memory maintenance ability signifies greater practicality in real-world applications such as autonomus driving.
> >
> > **Revision of Q4 (summarized in Summary 1.1, 3.2 and 3.4):**
> > 1. We further highlight importance of solving memory maintenance.
> >     > Revision in paper: line 30-37, Section 1
> > 2. We revise the description of memory models used in our comparison experiments, emphsize their better memory update capabilities.
> >     > Revision in paper: line 358-360, Section 5
> > 3. We Add limitations discussion of efficient memory update.
> >     > Revision in paper: line 1154-1158, Appendix G
> >
> > ---
> > ### Q5: Discarding invalid runs of our method might result in unfair comparison.
> >
> > - This process was applied only to PyBullet tasks, which require efficient memory updating and are not our primary focus. Performance in these environments serves as reference only, demonstrating capability but not core contribution.
> >
> > **Revision of Q5 (summarized in Summary 3.1):**
> > 1. We add some sentences explaining that short-term memory tasks needs memory update more than memory maintenance, and is therefore not our focus.
> >     > Revision in paper: line 482-484, Section 5.2
> > 2. If necessary, these results can be removed, as they do not affect our main contributions.

---

> > > ### Comment · Reviewer_gBTK · 2025-11-27
> > >
> > > The authors' responses address most of my concerns, however, I remain concerned about the limited scope and applicability of the proposed method, based on these considerations, I am raising my score from 2 to 4 (marginally below the acceptance threshold).

---

### Official Review · Reviewer_M6hE · 2025-11-01

**Soundness:** 3
**Presentation:** 3
**Contribution:** 2
**Rating:** 4
**Confidence:** 3

**Summary:**

In this paper, the authors proposed a new recurrent cell which uses discrete gating instead of continuous gating. The authors argued that this gating mechanism is inspired by spiking neurons and the reason why discrete gating is better than continuous gating function due to having large surrogate gradients. Continuous gating mechanisms like sigmoid function which can have small gradient values and can lead to vanishing gradients. The authors evaluated their mechanism using various memory and distraction tasks.

**Strengths:**

1. The writing of the paper is clear, and the figures are easy to follow.
2. The mathematical equations in the main paper are also well presented and improved the readability.
3. Results in Passive Visual Match is clear and understandable.
4. Figure 3 and 4 also produced convincing results.

**Weaknesses:**

1. Having Figure 3, Figure 4 and Table 1 together at the same spot makes it feel extremely crowded. Best to separate them the figures and tables.
2. Why are results in the RepeatPrevious environment so much better than the baselines in the medium and hard category?
3. The proposed method did not do well in the POPGym benchmark. Can you authors give some insights on why? Also, which methods are considered state of the art?

**Questions:**

1. What is the memory consumption with respect to memory size compared to other models? Is it more memory efficient?

---

> ### Author Response · Authors · 2025-11-21
>
> We sincerely thank the reviewers for taking the time to read our manuscript and provide valuable feedback. We have organized these comments and provided responses and corresponding revisions accordingly. In the updated PDF, the revisions have been highlighted.
>
> Below are our responses to each specific point:
>
> ---
> ### W1: Having Figure 3, Figure 4 and Table 1 together at the same spot makes it feel extremely crowded.
>
> - Thank you for the advice. We agree that layout have a significant impact on the overall presentation of the article.
>
> **Revision of W1 (summarized in Summary 2.2):**
> 1. We have optimized the layout of those figures and tables. Figure 4 is now at the top of Page 8 while Table 1 and Figure 5 lies at the bottom of Page 8. Table 2 is put to Page 9.
>     > Revision in paper: Figure 4, Table 1 and Figure 5 on Page 8, Table 2 on Page 9
>
> ---
>
> ### W2: Why are results in the RepeatPrevious environment so much better than the baselines in the medium and hard category?
>
> - The RepeatPrevious environment requires the agent to reproduce outputs from several steps earlier. In the easy setting, the gap is short (k=4), while in the medium and hard settings, the gap is longer (k=32 and k=64).
>
> - This makes the task more demanding in terms of memory maintenance. Unlike tasks that require frequent memory updating, RepeatPrevious primarily tests the ability to maintain memory over time.
>
> - Our method is specifically designed to handle memory maintenance efficiently, which explains its superior performance in the medium and hard categories.
>
> **Revision of W2 (summarized in Summary 3.1):**
> 1. We clarified the distinction between RepeatPrevious Easy, Medium, and Hard settings in the experimental section.
>     > Revision in paper: line 463-464, Section 5.2
> 2. We emphasized the observed phenomenon and explained why RepeatPrevious highlights differences in difficulty and the need for stronger maintenance.
>     > Revision in paper: line 357-360, Section 5
> 3. In the problem definition and method description, we stressed that our approach focuses on efficient memory maintenance, which is the central topic of this paper.
>     > Revision in paper: line 153-156, 160-161, 190-191, Section 3 and line 292-294, line 300-301, Section 4.1
>
> ---
>
> ### W3: The proposed method did not do well in the POPGym benchmark. Can you authors give some insights on why? Also, which methods are considered state of the art?
>
> - The three POPGym environments (Autoencode, Battleship, Concentration) differ from pure long‑term memory maintenance tasks. They require stronger memory updating abilities. For example, Autoencode requires the model to output a reversed sequence, which demands continuous updating of the current output progress.
>
> - Our method does not explicitly enhance memory updating, so its advantage is less pronounced in these tasks. More specifically, our recurrent update mechanism is relatively simple. We use a minimal design based on the Simple Recurrent Unit, with only slight modifications for parallel training and memory maintenance. This simplicity may weaken our memory updating ability.
>
> - SHM is considered the state‑of‑the‑art method. It employs a large memory matrix (128×128), allowing more information to be stored without pre‑filtering irrelevant data through gating. This provides stronger memory updating and maintenance. Although our method does not use a matrix memory space, it still achieves performance close to SHM in some tasks, demonstrating its effectiveness.
>
> **Revision of W3 (summarized in Summary 3.1, 3.2 and 3.4):**
> 1. We added detailed descriptions of Autoencode, Battleship, and Concentration to highlight their differences from our problem setting.
>     > line 361-364, 472-475, Section 5
> 2. We revised the analysis of SHM's memory capacity, emphasizing how larger capacity benefits memory updating.
>     > Revision in paper: line 362-365, 477, Section 5
> 3. In experimental section, we add discussions about why our method didn't do well on those tasks.
>     > Revision in paper: line 358-360, Section 5
> 4. In the limitations section, we pointed out the weakness of our method in memory updating.
>     > Revision in paper: line 1154-1158, Appendix G
> 5. We clarified in the introduction that our focus is on memory maintenance rather than updating.
>     > Revision in paper: line 37-39, Section 1

---

> > ### Author Response · Authors · 2025-11-21
> >
> > ### Q1: What is the memory consumption with respect to memory size compared to other models? Is it more memory efficient?
> >
> > - Like other linear RNNs, our method’s memory consumption increases linearly with memory size. Because the use of the parallel scan algorithm.
> >
> > - In practice, our method has a slightly larger constant in space and time complexity. This constant depends on factors such as the implementation of parallel scan, the number of scan calls, and the size of input tensors. Compared to LRU (which uses a similar PyTorch implementation), our method requires two scan operations per forward pass (one for the recurrent unit and one for the spiking neuron). However, when keeping the recurrent state size consistent, the input tensor size per scan in our method is half that of LRU. Overall, our constant is slightly larger than LRU’s but remains very close.
> >
> > - Experimental results confirm this analysis. In the Passive Visual Match experiment, our method consumed 1948 MB, while LRU consumed 1910 MB, showing comparable efficiency.
> >
> > **Revision of Q1 (summarized in Summary 4.4):**
> > 1. We added a section in the appendix discussing theoretical time complexity and the role of parallel scan.
> >     > Revision in paper: line 1026-1053, Appendix E.6
> > 2. We reported computational efficiency results.
> >     > Revision in paper: Table 6 on Page 20

---

> > ### Comment · Reviewer_M6hE · 2025-11-27
> >
> > I thank the authors for the detailed responses. They have addressed my concerns and I will increase my score from 4 to 6.

---

### Author Response · Authors · 2025-11-21
**Summary of Revisions**

Dear Reviewers,

Thank you for your insightful comments and constructive suggestions. We have thoroughly revised our manuscript to address your feedback.

Below is a **summary** of the key changes made:

---

### 1. **Highlighted Contribution**
1. Narrowed down the problem definition. We now clearly differentiate between **memory maintenance** and **memory update**, underscoring that our work specifically targets **long-term memory maintenance**. The practical importance of this challenge is further emphasized in **Section 1 and Section 3**.
2. Reorganized Section 4 to emphasize the analysis of discrete gating for memory maintenance **(Section 4.1)** and its implementation **(Section 4.2)**, while reducing the focus on the design of the recurrent cell.

---

### 2. **Improved Paper Structure and Readability**
1. Moved background on POMDPs (previously in Section 2) and detailed cell implementations (previously in Section 4) to **Appendix A and Appendix B**, allowing greater focus on our core contributions.
2. Simplified section titles and split multi-part figures (e.g., **Figure 1**) for better clarity.
3. Reorganized all figures and tables in **Section 5** to enhance visual flow and comprehension.

---

### 3. **Enhanced Analysis of Method Performance**
1. Clearly outlined task types where our method excels (e.g., Passive Visual Match, Repeat Previous) versus those requiring stronger memory update capabilities (e.g., general and short-term memory task) in **Section 5**.
2. Added **Table 1** to compare memory modeling complexity across methods, explaining that our focus on maintenance leads to simpler update mechanisms.
3. Provided targeted result analysis in **Section 5**, linking performance differences to memory maintenance efficiency.
4. Discuss our method's limitation of memory update **(Appendix G)**.

---

### 4. **Expanded Experimental Validation**
1. Added comparisons with **Mamba**, revealing its limitations in long-term memory settings **(Appendix E.5)**.
2. Conducted **ablation studies** to verify the roles of discrete gating, surrogate gradients, and stochastic thresholds **(Section 5.3)**.
3. Included results on the **12-AX** and **T-Maze** tasks in **Appendix E.2 and E.4**, demonstrating broader applicability.
4. Provided **time complexity and memory consumption analysis** in **Appendix E.6**, confirming comparable efficiency to linear RNNs with a modest constant-factor overhead.

---

### 5. **Clarified Key Design Choices**
1. Justified the **absence of a reset mechanism** to preserve historical context and support parallel scanning. **(Section 4.2)**
2. Clarified that **discrete gating improves memory maintenance** by enabling strong noise filtering without gradient vanishing. **(Section 4.1)**

---

We believe these revisions have substantially strengthened the paper. We are open to any further suggestions and appreciate your valuable time and guidance.

Sincerely,
The Authors

---

### Comment · Area_Chair_Hv9Y · 2025-11-24

Dear Reviewers,

The authors have responded to your reviews. Please review and respond to their comments who have not yet done so.

Best, Your AC

---

### Author Response · Authors · 2025-12-02
**Summary of the discussion phase (PART 1/2, first revision)**

Dear AC,

We thank all reviewers for their constructive feedback on our paper. We especially appreciate your work as the AC during this unique time, which we understand involved significant extra effort and diligence to ensure the quality of the review process. Hope the summary below alleviates some of the workload in the process.

We made **two round**s of revisions during the rebuttal process. After our first-round revision, two reviewers increase their scores and one reviewer maintain his score of 8:

- Reviewer M6hE: 4 → 6 (responded at ~1h after the identity leak)
- Reviewer gBTK: 2 → 4 (responded **before** the identity leak)
- Reviewer ELid: 8 → 8 (responded **before** the identity leak)
- Reviewer oAXs: 4 → 4 (without any response)

We confirm that **no attempt from us was made to locate or contact any reviewer or AC, and no other improper outreach occurred.**

---
### Revision BEFORE the leak

We have provided **point‑by‑point** responses to **all reviewers**. The summary of key concerns and our answers are as follows:

**Common Question from Reviewer M6hE (Weakness 3), gBTK (Weakness 3, Question 2), and oAXs (Weakness 2):**

**Q:** Insights into why our method underperforms on general memory tasks and short-term memory tasks.

**A:** We explained that:
- Our focus is memory maintenance rather than updating (problem clarification at line 37-39),
- which leads to superior performance in RepeatPrevious (task description in line 453-457) but weaker results in POPGym tasks (task description found at line 477-480, 497) and short‑term tasks (task description found at line 915-916), experimental results at line 444-449, 486-496 and 919-932.
- We also highlighted SHM as the state‑of‑the‑art baseline and discussed differences in memory updating ability (line 362-365).

**Reviewer M6hE:**

**Q:** The effect of memory length in RepeatPrevious on different methods (Weakness 2).

**A:** We clarified the distinction between RepeatPrevious Easy, Medium, and Hard settings (line 453-457), emphasizing that our method excels in memory maintenance rather than updating (line 344-346).

**Reviewer gBTK:**

**Q:** Comparison with Mamba and other SOTA methods (W1, Q3).

**A:** We added experiments comparing our method with Mamba, showing that Mamba underperforms in Passive Visual Match and T‑Maze (line 1020-1024, 1064-1070).

**Q:** Motivation for using spiking neurons (Q1).

**A:** We clarified the theoretical motivation for discrete gating with spiking neurons (line 291-292, 298-299), supported by ablation studies (line 509-518).

**Q:** Fairness of experimental design on short-term memory tasks (Q5).

**A:** We clarify the purpose of conducting these experiments and emphasize that they do not impact fairness. (line 912-915).

**Reviewer ELid:**

**Q:** Request for more ablation experiments (W1).

**A:** We added ablation studies on surrogate gradient functions and stochastic thresholding, showing robustness of our design (line 519-531).

**Reviewer oAXs:**

**Q:** Request for experiments on more classical memory tasks (W3, Q4).

**A:** We added experiments on T‑Maze and 12‑AX tasks, demonstrating competitive performance (line 863, 888-893, 949-970).

**Q:** More insights into design choices, including reset and gating (Q2, Q3).

**A:** We summarized when discrete gating is advantageous (long noise phases, line 68-69) and when it is limited (tasks requiring complex memory updating, line 1212-1216).

The **specific modifications** are marked in **yellow** and summarized at our **"Summary of Revisions"** (our general comments in first-round response).

---

> ### Author Response · Authors · 2025-12-02
> **Summary of the discussion phase (PART 2/2, second revision)**
>
> ### Revision AFTER the leak
>
> After the first round of responses, three reviewers replied positively, among which:
>
> - Reviewer ELid and gBTK replied before the leak. ELid maintains his score as 8, and gBTK raises his score from 2 to 4 with concerns regarding **our method's scope and applicability**.
>
> - Reviewer M6hE increased score from 4 to 6 about 1h after the leak.
>
> - Reviewer oAXs did not response.
>
> We conduct additional experiments with a **different learning manner (i.e., supervised learning)** and **more realistic environments** to address gBTK's concern.
> - **Supervised learning task: the Adding Problem.** According to our analysis in Section 3.1, our core contribution of gating mechanism is also expected to be applicable to supervised learning tasks. Following Reviewer oAXs’s suggestion, we evaluated it on the classic Adding Problem. Experiments show that compared to SOTA gated RNNs, such as GRU and FFM, our approach converges the fastest on long sequences, with its advantage growing as sequence length increases, demonstrating strong long-term memory maintenance.
>
> - **Tasks with more realistic environments: Minecraft Maze.** Minecraft Maze is a set of tasks designed based on Malmo, which is used to evaluate the performance of long-term memory agents in high-dimensional observation spaces. We test on the Minecraft Maze task (42×42 image input). Compared with SOTA RNNs such as FFM and Mamba, our method achieves the highest return within fewest timesteps.
>
>
> The **specific modifications** are marked in **pink** and summarized below:
> - The results of the Adding Problem are now included in the second-to-last block of Section 5.2 to validate our core contribution of gating mechanism in supervised learning. Extended sequence-length tests and task details are provided in Appendix F.2.
> - To meet length constraints, the analysis of short-term memory tasks has been moved to Appendix E.2.
> - Due to space limitations, the results of the Minecraft Maze experiment have been placed in Appendix E.3.
> - Descriptions of newly added experiments: the adding problem (line 1117-1122) and Minecraft Maze (line 894-900). Their hyperparameter details are provided in Appendix F.3 (line 1189-1196).
>
> ---
> We regret that the information leak has increased your workload, and we deeply appreciate your efforts in addressing this matter. To help streamline the process, we have carefully structured our discussion. We are confident that the revisions and additional experiments have further strengthened the paper. Finally, we must reiterate that we have never accessed, seeked out, or otherwise obtained the disclosed information, nor engaged in any other non-compliant or unauthorized actions during rebuttal.
>
> Sincerely,
>
> The Authors

---

### Meta-Review · Area_Chair_V3vW · 2025-12-08

**Summary:**

The paper proposes a discrete gating mechanism based on spiking neurons for linear RNNs, and empirically shows benefits on a set of long-horizon, noisy POMDP tasks. While the idea of discrete gating for long-term memory is interesting, the current submission does not convincingly justify why the specific “spiking neuron” parameterization is preferable to simpler hard-gating alternatives (e.g., sigmoid + Gumbel/straight-through Bernoulli), nor does it include such baselines. Combined with the already-raised concerns about narrow task coverage and limited impact beyond this specific setting, I do not think the contribution meets the bar for acceptance.

**Reviewer Concerns:**

I agree with the reviewers that the theoretical analysis supports the value of discrete gating on the recurrent dynamics, but it does not provide a principled argument that the proposed spiking neuron is superior to other standard hard-gate implementations; the benefit of “spiking” per se remains under-motivated. Moreover, from an architectural point of view, a sigmoid-based hard gate with Gumbel-Softmax / straight-through training is a trivial baseline that would inherit the same structural advantages (hard skip, noise not back-propagating) the theory relies on; the absence of such a baseline, or of any discussion of why it would be inadequate, is a significant missing comparison. Earlier concerns about the method’s narrow scope (tasks with explicit signal–noise phases, RL mainly as a sequence-model evaluation setting) remain largely valid, so the paper currently reads more as one particular instantiation of hard gating than as a robust argument for spiking neurons as uniquely well-justified.

**Reviewer Scores:**

Given the above, I side with the more skeptical reviewers: I believe the low and mid-range scores would likely stay unchanged or only slightly improve if this specific “spiking vs. trivial hard gate” issue had been explicitly discussed. The most positive review focuses on the general idea of discrete gating and SNR analysis, but does not address the lack of comparison to equally capable hard-gating baselines. Overall, I view the effective consensus as “borderline with substantial unresolved concerns” and therefore recommend rejection.

---

### Decision · Program_Chairs · 2026-01-26

Reject